# Overcoming Catastrophic Forgetting in Incremental Few-Shot Learning by Finding Flat Minima

**Guangyuan Shi,**[*] **Jiaxin Chen,**[*] **Wenlong Zhang, Li-Ming Zhan, Xiao-Ming Wu**[†]
Department of Computing
The Hong Kong Polytechnic University
{guang-yuan.shi, jiax.chen, wenlong.zhang, lmzhan.zhan}@connect.polyu.hk
xiao-ming.wu@polyu.edu.hk

## Abstract

This paper considers incremental few-shot learning, which requires a model to continually recognize new categories with only a few examples provided. Our study shows that existing methods severely suffer from catastrophic forgetting, a well-known problem in incremental learning, which is aggravated due to data scarcity and imbalance in the few-shot setting. Our analysis further suggests that to prevent catastrophic forgetting, actions need to be taken in the primitive stage – the training of base classes instead of later few-shot learning sessions. Therefore, we propose to search for flat local minima of the base training objective function and then fine-tune the model parameters within the flat region on new tasks. In this way, the model can efficiently learn new classes while preserving the old ones. Comprehensive experimental results demonstrate that our approach outperforms all prior state-of-the-art methods and is very close to the approximate upper bound. The source code is available at `https://github.com/moukamisama/F2M`.

## 1 Introduction

**Why study incremental few-shot learning?** Incremental learning enables a model to continually learn new concepts from new data without forgetting previously learned knowledge. Rooted from real-world applications, this topic has attracted a significant amount of interest in recent years [5, 30, 31, 40, 26]. Incremental learning assumes sufficient training data is provided for new classes, which is impractical in many application scenarios, especially when the new classes are rare categories which are costly or difficult to collect. This motivates the study of incremental few-shot learning, a more difficult paradigm that aims to continually learn new tasks with only a few examples.

**Challenges.** The major challenge for incremental learning is *catastrophic forgetting* [14, 28, 35], which refers to the drastic performance drop on previous tasks after learning new tasks. This phenomenon is caused by the inaccessibility to previous data while learning on new data. Catastrophic forgetting presents a bigger challenge for incremental few-shot learning. Due to the small amount of training data in new tasks, the model tends to severely overfit on new classes while quickly forgetting old classes, resulting in catastrophic performance.

**Current research.** The study of incremental few-shot learning has just started [47, 41, 60, 9, 8, 34, 59]. Current works mainly borrow ideas from research in incremental learning to overcome the forgetting problem, by enforcing strong constraints on model parameters to penalize the changes of parameters [34, 28, 56], or by saving a small amount of exemplars from old classes and adding constraints on the exemplars to avoid forgetting [40, 20, 4]. However, in our empirical study, we find

---

[*]Equal contribution
[†]Corresponding author

35th Conference on Neural Information Processing Systems (NeurIPS 2021).

that an intransigent model that only trains on base classes and does not tune on new tasks consistently outperforms state-of-the-art methods, including a joint-training method [47] that uses all encountered data for training and hence suffers from severe data imbalance. This observation motivates us to address this harsh problem from a different angle.

**Our solution.** Unlike existing solutions that try to overcome the catastrophic forgetting problem during the process of learning new tasks, we adopt a different approach by considering this issue during the training of base classes. Specifically, we propose to search for flat local minima of the base training objective function. For any parameter vector in the flat region around the minima, the loss is small, and the base classes are supposed to be well separated. The flat local minima can be found by adding random noise to the model parameters for multiple times and jointly optimizing multiple loss functions. During the following incremental few-shot learning stage, we fine-tune the model parameters within the flat region, which can be achieved by clamping the parameters after updating them on few-shot tasks. In this way, the model can efficiently learn new classes while preserving the old ones. Our key contributions are summarized as follows:

- We conduct a comprehensive empirical study on existing incremental few-shot learning methods and discover that a simple baseline model that only trains on base classes outperforms state-of-the-art methods, which demonstrates the severity of catastrophic forgetting.

- We propose a novel approach for incremental few-shot learning by addressing the catastrophic forgetting problem in the primitive stage. Through finding the flat minima region during training on base classes and fine-tuning within the region while learning on new tasks, our model can overcome catastrophic forgetting and avoid overfitting.

- Comprehensive experimental results on CIFAR-100, *mini*ImageNet, and CUB-200-2011 show that our approach outperforms all state-of-the-art methods and achieves performance that is very close to the approximate upper bound.

## 2 Related Work

**Few-shot learning** aims to learn to generalize to new categories with a few labeled samples in each class. Current few-shot methods mainly include optimization-based methods [12, 23, 32, 39, 45, 46, 55] and metric-based methods [13, 19, 44, 49, 52, 58, 57, 53]. Optimization-based methods can achieve fast adaptation to new tasks with limited samples by learning a specific optimization algorithm. Metric-based approaches exploit different distance metrics such as L2 distance [44], cosine similarity [49], and DeepEMD [58] in the learned metric/embedding space to measure the similarity between samples. Recently, Tian *et al.* [48] find that standard supervised training can learn a good metric space for unseen classes, which echoes with our observation on the proposed baseline model in Sec. 3.

**Incremental learning** focuses on the challenging problem of continually learning to recognize new classes in new coming data without forgetting old classes [6, 7, 10, 51]. Previous research mainly includes multi-class incremental learning [4, 38, 22, 33, 54, 51] and multi-task incremental learning [21, 31, 42]. To overcome the catastrophic forgetting problem, some attempts propose to impose strong constraints on model parameters by penalizing the changes of parameters [28, 1]. Other attempts try to enforce constraints on the exemplars of old classes by restricting the output logits [40] or penalizing the changes of embedding angles [20]. In this work, our empirical study shows that imposing strong constraints on the arriving new classes may not be a promising way to tackle incremental few-shot learning, due to the scarcity of training data for new classes.

**Incremental few-shot learning** [47, 41, 60, 9, 8] aims to incrementally learn from very few samples. TOPCI [47] proposes a neural gas network to learn and preserve the topology of the feature manifold formed by different classes. FSLL [34] only selects few model parameters for incremental learning and ensures the parameters are close to the optimal ones. To overcome catastrophic forgetting, IDLVQC [8] imposes constraints on the saved exemplars of each class by restricting the embedding drift, and Zhang *et al.* [59] propose to fix the embedding network for incremental learning. Similar to the finding of Zhang *et al.*, we also discover that an intransigent model that simply does not adapt to new tasks can outperform prior state-of-the-art methods.

**Robust optimization.** It has been found that flat local minima leads to better generalization capabilities than sharp minima in the sense that a flat minimizer is more robust when the test loss is shifted

due to random perturbations [18, 17, 24]. A substantial body of methods [2, 37, 11, 15] have been proposed to optimize neural networks towards flat local minima. In this paper, we show that for incremental few-shot learning, finding flat minima in the base session and tuning the model within the flat region on new tasks can significantly mitigate catastrophic forgetting.

# 3 Severity of Catastrophic Forgetting in Incremental Few-Shot Learning

## 3.1 Problem Statement

Incremental few-shot learning (IFL) aims to continually learn to recognize new classes with only few examples. Similar to incremental learning (IL), an IFL model is trained by a sequence of training sessions $\{\mathcal{D}^1, \cdots, \mathcal{D}^t\}$, where $\mathcal{D}^t = \{z_i = (x_i^t, y_i^t)\}_i$ is the training data of session $t$ and $x_i^t$ is an example of class $y_i^t \in \mathcal{C}^t$ (the class set of session $t$). In IFL, the base session $\mathcal{D}^1$ usually contains a large number of classes with sufficient training data for each class, while the following sessions ($t \geq 2$) only have a small number of classes with few training samples per class, e.g., $\mathcal{D}^t$ is often presented as an $N$-way $K$-shot task with small $N$ and $K$. The key difference between IL and IFL is, for IL, sufficient training data is provided in each session. Similar to IL, in each training session $t$ of IFL, the model has only access to the training data $\mathcal{D}^t$ and possibly a small amount of saved exemplars from previous sessions. When the training of session $t$ is completed, the model is evaluated on test samples from all encountered classes $\mathcal{C} = \bigcup_{i=1}^t \mathcal{C}^i$, where it is assumed that there is no overlap between the classes of different sessions, i.e., $\forall i, j$ and $i \neq j, \mathcal{C}^i \bigcap \mathcal{C}^j = \emptyset$.

**Catastrophic forgetting.** IFL is undoubtedly a more challenging problem than IL due to the data scarcity setting. IL suffers from catastrophic forgetting, a well-known phenomenon and long-standing issue, which refers to the drastic drop in test performance on previous (old) classes, caused by the inaccessibility of old data in the current training session. Unfortunately, catastrophic forgetting is an even bigger issue for IFL, because data scarcity makes it difficult to adapt well to new tasks and learn new concepts, while the adaptation process could easily lead to the forgetting of base classes. In the following, we illustrate this point by evaluating a simple baseline model for IFL.

## 3.2 A Simple Baseline Model for IFL

We consider an intransigent model that simply does not adapt to new tasks. Particularly, the model only needs to be trained in the base session $\mathcal{D}^1$ and is directly used for inference in all sessions.

**Training ($t = 1$).** We train a feature extractor $f$ parameterized by $\phi$ with a fully-connected layer as classifier by minimizing the standard cross-entropy loss using the training examples of $\mathcal{D}^1$. The feature extractor $f$ is *fixed* for the following sessions ($t \geq 2$) without any fine-tuning on new classes.

**Inference (test).** In each session, the inference is conducted by a simple nearest class mean (NCM) classification algorithm [36]. Specifically, all the training and test samples are mapped to the embedding space of the feature extractor $f$, and Euclidean distance $d(\cdot, \cdot)$ is used to measure the similarity between them. The classifier is given by

$$c_k^\star = \operatorname*{argmin}_{c \in \mathcal{C}} d(f(x; \phi), p_c), \text{ where } p_c = \frac{1}{N_c} \sum_i \mathbb{1}(y_i = c) f(x_i; \phi), \tag{1}$$

where $\mathcal{C}$ denotes all the encountered classes, $p_c$ refers to the prototype of class $c$ (the mean vector of all the training samples of class $c$ in the embedding space), and $N_c$ denotes the number of the training images of class $c$. Note that we *save* the prototypes of all classes in $\mathcal{C}^t$ for later evaluation.

**The baseline model outperforms state-of-the-art IFL and IL methods.** We compare the above baseline model against state-of-the-art IFL methods including FSLL [34], IDLVQC [8] and TOPIC [47], IL methods including Rebalance [20] and iCarl [40], and a joint-training method that uses all previously seen data including the base and the following few-shot tasks for training, for IFL. The performance is evaluated on miniImageNet, CIFAR-100, and CUB-200. We tune the methods re-implemented by us to the best performance. For the other methods, we use the results reported in the original papers. The experimental details are provided in Sec. 5. As shown in Fig. 1, the baseline model consistently outperforms all the compared methods including the joint-training

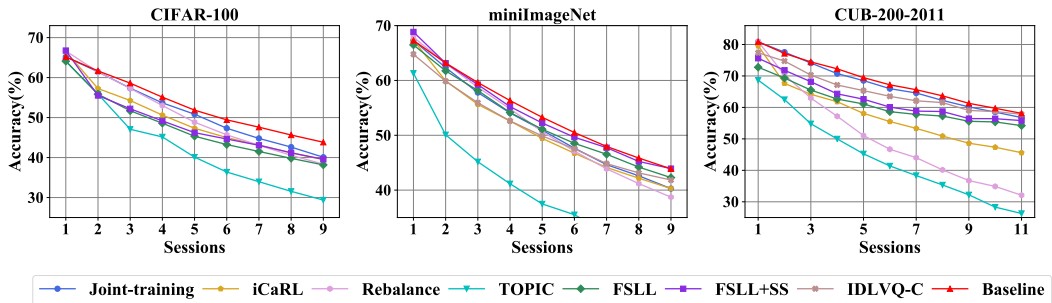

Figure 1: Comparison of the proposed baseline model with state-of-the-art IFL and IL methods and the joint-training method.The baseline model outperforms all the other methods.

method (which suffers from severe data imbalance) on every dataset[3]. The fact that an intransigent model performs best suggests that

- For IFL, preserving the old (base classes) may be more critical than adapting to the new. Due to data scarcity, the performance gain on new classes is limited and cannot make up for the significant performance drop on base classes.

- Prior works [47, 8, 34, 20, 40] that enforce strong constraints on model parameters or exemplars during fine-tuning on new classes cannot effectively prevent catastrophic forgetting in IFL, indicating that actions may need to be taken in the base training stage.

## 4 Overcoming Catastrophic Forgetting in IFL by Finding Flat Minima

The goal of IFL is to preserve the old while adapting to the new efficiently. The results and analysis in Sec. 3 suggest that it might be "a bit late" to try to prevent catastrophic forgetting in the few-shot learning sessions ($t \geq 2$), which motivates us to consider this problem in the base training session.

**Overview of our approach.** To overcome catastrophic forgetting in IFL, we propose to find a $b$-flat ($b > 0$) local minima $\theta^\star$ of the base training objective function and then fine-tune the model within the flat region in later few-shot learning sessions. Specifically, for any parameter vector $\theta$ in the flat region, i.e., $\theta^\star - b \preceq \theta \preceq \theta^\star + b$, the risk (loss) of the base classes is minimized such that the classes are well separated in the embedding space of $f_\theta$. In the later incremental few-shot learning sessions ($t \geq 2$), we fine-tune the model parameters within this region to learn new classes, i.e., to find

$$\theta' = \arg \min_\theta \sum_{z \in \mathcal{D}^t} \mathcal{L}(z; \theta), \ \ \text{s.t.} \ \ \theta^\star - b \preceq \theta \preceq \ \theta^\star + b.$$

As such, the fine-tuned model $\theta'$ can adapt to new classes while preserving the old ones. Also, due to the nature of few-shot learning, to avoid excessive training and overfitting, it suffices to tune the model in a relatively small region. A graphical illustration of our approach and prior arts, as well as the notions of sharp minima and flat minima, are presented in Fig. 2.

### 4.1 Searching for Flat Local Minima in the Base Training Stage

A formal definition of $b$-flat local minima is given as follows.

**Definition 1** ($b$-Flat Local Minima). *Given a real-valued objective function $\mathcal{L}(z; \theta)$, for any $b > 0$, $\theta^\star$ is a b-flat local minima of $\mathcal{L}(z; \theta)$, if the following conditions are satisfied.*

- *Condition 1: $\mathcal{L}(z; \theta^\star) = \mathcal{L}(z; \theta^\star + \epsilon)$, where $-\mathbf{b} \preceq \epsilon \preceq \mathbf{b}$ and $\mathbf{b}_i = b$.*

- *Condition 2: there exist $\mathbf{c}_1 \prec \theta^\star - b$ and $\mathbf{c}_2 \succ \theta^\star + b$, s.t. $\mathcal{L}(z; \theta) > \mathcal{L}(z; \theta^\star)$, where $\mathbf{c}_1 \prec \theta \prec \theta^\star - b$ and $\mathcal{L}(z; \theta^\star) < \mathcal{L}(z; \theta)$, where $\theta^\star + b \prec \theta \prec \mathbf{c}_2$.*

---

[3]We notice that a similar observation is made in a newly released paper [59].

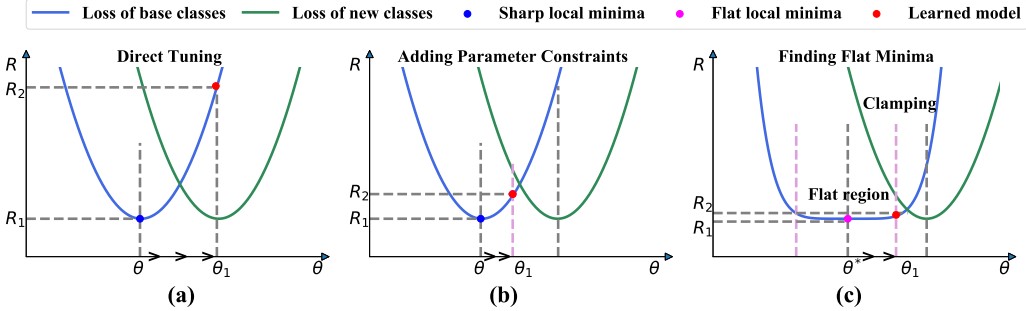

Figure 2: Illustration of our approach and existing solutions. $\rightarrow$ indicates the incremental learning steps on new classes. $R_1$ and $R_2$ respectively denote the loss of base classes before and after minimizing the loss of new classes. (a) SGD finds sharp minima in the base training. Directly tuning the model on new classes will result in a severe performance drop on base classes. (b) Enforcing strong constraints on parameters by penalizing parameter changes [1, 28, 34] may still lead to a significant performance drop on base classes. (c) Finding flat local minima of base classes and clamping the parameters after trained on new classes to make them fall within the flat region can effectively mitigate catastrophic forgetting.

In practice, it is hard to find the flat local minima that strictly satisfies the above definition, which may not even exist. Hence, our goal is to find an approximately flat local minima of the base training objective function. To this end, we propose to add some small random noise to the model parameters. The noise can be added for multiple times to obtain similar but different loss functions, which will be optimized together to locate the flat minima region. The intuition is clear – the parameter vectors around the flat local minima also have small function values.

To formally state the idea, we assume that the model is parameterized by $\theta = \{\phi, \psi\}$, where $\phi$ denotes the parameters of the embedding network and $\psi$ denotes the parameters of the classifier. $z$ denotes a labelled training sample. Denote the loss function by $\mathcal{L}$: $\mathbb{R}^{d_z} \to \mathbb{R}$. Our target is to minimize the expected loss function $R$: $\mathbb{R}^d \to \mathbb{R}$ w.r.t. the joint distribution of data $z$ and noise $\epsilon$, i.e.,

$$R(\theta) = \int_{\mathbb{R}^{d_\epsilon}} \int_{\mathbb{R}^{d_z}} \mathcal{L}(z; \phi + \epsilon, \psi) \, dP(z)dP(\epsilon) = \mathbb{E}[\mathcal{L}(z; \phi + \epsilon, \psi)], \tag{2}$$

where $P(z)$ is the data distribution and $P(\epsilon)$ is the noise distribution, and $z$ and $\epsilon$ are independent. Since it is impossible to minimize the expected loss, we minimize its estimation, the empirical loss, which is given by

$$\mathcal{L}(\theta) = \frac{1}{M} \sum_{j=1}^{M} \mathcal{L}_{\text{base}}(z; \phi + \epsilon_j, \psi), \text{ where} \tag{3}$$

$$\mathcal{L}_{\text{base}}(z; \phi + \epsilon_j, \psi) = \frac{1}{|\mathcal{D}^1|} \sum_{z \in \mathcal{D}^1} \mathcal{L}_{ce}(z; \phi + \epsilon_j, \psi) + \lambda \frac{1}{|\mathcal{C}^1|} \sum_{c \in \mathcal{C}^1} \|p_c - p_c^*\|_2^2, \tag{4}$$

where $\epsilon_j$ is a noise vector sampled from $P(\epsilon)$, $M$ is the sampling times, $\mathcal{L}_{ce}(z; \phi + \epsilon_j, \psi)$ refers to the cross-entropy loss of a training sample $z$, and $p_c$ and $p_c^*$ are the class prototypes before and after injecting noise respectively. The first term of $\mathcal{L}_{base}$ is designed to find the flat region where the parameters $\phi$ of the embedding network can well separate the base classes. The second term enforces the class prototypes fixed within such region, which is designed to solve the prototype drift problem [54, 8] (the class prototypes change after updating the network) in later incremental learning sessions such that the saved base class prototypes can be directly used for evaluation in later sessions.

## 4.2 Incremental Few-shot Learning within the Flat Region

In the incremental few-shot learning sessions ($t \geq 2$), we fine-tune the parameters $\phi$ of the embedding network *within the flat region* to learn new classes. It is worth noting that while the flat region might be relatively small, it is enough for incremental few-shot learning. Because only few training samples are provided for each new class, to prevent overfitting in few-shot learning, excessive training should be avoided and only a small number of update iterations should be applied.

**Algorithm 1:** F2M

---

**Input:** the flat region bound $b$, randomly initialized $\theta = \{\phi, \psi\}$, the step sizes $\alpha$ and $\beta$.

// `Training over base classes` $t = 1$

**for** epoch $k$ = 1,2,... **do**

    **for** $j$ = 1,2,..., $M$ **do**

        Sample a noise vector $\epsilon_j \sim P(\epsilon)$, s.t. $-\mathbf{b} \preceq \epsilon_j \preceq \mathbf{b}$;

        Add the noise to the parameters of the embedding network, i.e., $\theta = \{\phi + \epsilon_j, \psi\}$;

        Compute the base loss $\mathcal{L}_{base}$ with Eq. 4;

        Reset the parameters, i.e., $\theta = \{\phi, \psi\}$;

    **end**

    Update $\theta = \theta - \alpha \nabla \mathcal{L}(\theta)$ with the loss $\mathcal{L}$ defined in Eq. 3.

**end**

Normalize and save the prototype of each base class;

// `Incremental learning` $t \geq 2$

Combine the training data $\mathcal{D}^t$ and the exemplars saved in previous few-shot sessions $2 \leq t_e < t$;

**for** epoch $k$ = 1,2,... **do**

    Compute the metric-based classification loss $\mathcal{L}_m$ by Eq. 5;

    Update $\phi = \phi - \beta \nabla \mathcal{L}_m(z; \phi)$;

    Clamp the parameters $\phi$ to ensure they fall in the flat minima region;

**end**

Randomly select and save a few exemplars from the training data $\mathcal{D}^t$;

Normalize and save the prototype of each new class;

**Output:** Model parameters $\theta = \{\phi, \psi\}$.

---

We employ a metric-based classification algorithm with Euclidean distance to fine-tune the parameters. The loss function is defined as

$$\mathcal{L}_m(z; \phi) = -\sum_{z \in \mathcal{D}} \sum_{c \in \mathcal{C}} \mathbb{1}(y = c) \log\left(\frac{e^{-d(p_c, f(x; \phi))}}{\sum_{c_k \in \mathcal{C}} e^{-d(p_{c_k}, f(x; \phi))}}\right), z \quad (5)$$

where $d(\cdot, \cdot)$ denotes Euclidean distance, $p_c$ is the prototype of class $c$, $\mathcal{C} = \bigcup_{i=1}^t \mathcal{C}^i$ refers to all encountered classes, and $\mathcal{D} = \mathcal{D}^t \bigcup \mathcal{P}$ denotes the union of the current training data $\mathcal{D}^t$ and the exemplar set $\mathcal{P} = \{P_2, ..., P_{t-1}\}$, where $P_{t_e}(2 \leq t_e < t)$ is the set of saved exemplars in session $t_e$. Note that the prototypes of new classes are computed by Eq. 1, and those of base classes are saved in the base session. After updating the embedding network parameters, we clamp them to ensure that they fall within the flat region, i.e. $\phi^\star - b \preceq \phi \preceq \phi^\star + b$, where $\phi^\star$ denotes the optimal parameter vector learned in the base session. After fine-tuning, we evaluate the model using the nearest class mean classifier as in Eq. 1, with previously saved prototypes and newly computed ones. The whole training process is described in Algorithm 1. Note that to calibrate the estimates of the classifier, we normalize all prototypes to make those of base classes and those of new classes have the same norm.

## 4.3 Convergence Analysis

Our aim is to find a flat region within which all parameter vectors work well. We then minimize the expected loss w.r.t. the joint distribution of noise $\epsilon$ and data $z$. To approximate this expected loss, we sample from $P(\epsilon)$ for multiple times in each iteration and optimize the objective function using stochastic gradient descent (SGD). Here, we provide theoretical guarantees for our method. Given the non-convex loss function in Eq. 3, we prove the convergence of our proposed method. The proof idea is inspired by the convergence analysis of SGD [3, 27].

Formally, in each batch $k$, let $z_k$ denote the batch data, $\{\epsilon_j\}_{j=1}^M$ be the sampled noises, and $\alpha_k$ be the step size. In the base training session, we update the model parameters as follows:

$$\theta_{k+1} = \theta_k - \frac{\alpha_k}{M} \sum_{j=1}^M \nabla \mathcal{L}_{\text{base}}(z_k; \phi_k + \epsilon_j, \psi_k) = \theta_k - \frac{\alpha_k}{M} \sum_{j=1}^M g(z_k; \phi_k + \epsilon_j, \psi_k), \quad (6)$$

where $g(z_k; \phi_k + \epsilon_j, \psi_k) = \nabla \mathcal{L}_{\text{base}}(z_k; \phi_k + \epsilon_j, \psi_k)$ is the gradient. To formally analyze the convergence of our algorithm, we define the following assumptions.

**Assumption 4.1** (L-smooth risk function). *The expected loss function $R : \mathbb{R}^d \to \mathbb{R}$ (Eq. 2) is continuously differentiable and L-smooth with constant $L > 0$ such that*

$$\|\nabla R(\theta) - \nabla R(\theta')\|_2 \leq L\|\theta - \theta'|. \tag{7}$$

This assumption is significant for the convergence analysis of gradient-based optimization algorithms, since it limits how fast the gradient of the loss function can change w.r.t. the parameter vector.

**Assumption 4.2.** *The expected loss function satisfies the following conditions:*

- *Condition 1: $R$ is bounded below by a scalar $R^\star$, given the sequence of parameters $\{\theta_k\}$.*

- *Condition 2: For all $k \in \mathbb{N}$ and $j \in [1, M]$,*

$$\mathbb{E}_{z_k, \epsilon_j}[g(z_k; \phi_k + \epsilon_j, \psi_k)] = \nabla R(\theta_k). \tag{8}$$

- *Condition 3: There exist scalars $m_1 \geq 0$ and $m_2 \geq 0$, for all $k \in \mathbb{N}$ and $j \in [1, M]$,*

$$\mathbb{V}_{z_k, \epsilon_j}[g(z_k; \phi_k + \epsilon_j, \psi_k)] \leq m_1 + m_2\|\nabla R(\theta_k)\|_2^2. \tag{9}$$

$\mathbb{E}_{z_k, \epsilon_j}[\cdot]$ denotes the expectation w.r.t. the joint distribution of random variables $z_k$ and $\epsilon_j$, and $\mathbb{V}_{z_k, \epsilon_j}[\cdot]$ denotes the variance. Condition 1 ensures that the expected loss $R$ is bounded by a minimum value $R^\star$ during the updates, which is a natural and practical assumption. Condition 2 assumes that the gradient $g(z_k; \phi_k + \epsilon_j, \psi_k)$ is an unbiased estimate of $\nabla R(\theta_k)$. This is a strict assumption made to simplify the proof, but it can be easily relaxed to a general and easily-met condition that there exist $\mu_1 \geq \mu_2 > 0$ satisfying $\|\mathbb{E}_{z_k, \epsilon_j}[g(z_k; \phi_k + \epsilon_j, \psi_k)]\|_2 \leq \mu_1\|\nabla R(\theta_k)\|_2$ and $\nabla R(\theta_k)^T \mathbb{E}_{z_k, \epsilon_j}[g(z_k; \phi_k + \epsilon_j, \psi_k)] \geq \mu_2\|\nabla R(\theta_k)\|_2^2$. Therefore, the convergence can be proved in a similar way using the techniques presented in the Appendix. Condition 3 assumes the variance of the gradient $g(z_k; \phi_k + \epsilon_j, \psi_k)$ cannot be arbitrarily large, which is also reasonable in practice. To facilitate later analysis, similar to [43], we restrict the step sizes as follows.

**Assumption 4.3.** *The learning rates satisfy:*

$$\sum_{k=1}^{\infty} \alpha_k = \infty, \ \ \sum_{k=1}^{\infty} \alpha_k^2 < \infty. \tag{10}$$

This assumption can be easily met, since in practice the learning rate $\alpha_k$ is usually far less than 1 and decreases w.r.t. $k$. Based on the above assumptions, we can derive the following theorem.

**Theorem 4.1.** *Under assumptions 4.1, 4.2 and 4.3, we further assume that the risk function $R$ is twice differentiable, and that $\|\nabla R(\theta)\|_2^2$ is $L_2$-smooth with constant $L_2 > 0$, then we have*

$$\lim_{k \to \infty} \mathbb{E}[\|\nabla R(\theta_k)\|_2^2] = 0. \tag{11}$$

This theorem establishes the convergence of our algorithm. The proof is provided in Appendix A.1.

## 5 Experiments

In this section, we empirically evaluate our proposed method for incremental few-shot learning and demonstrate its effectiveness by comparison with state-of-the-art methods.

### 5.1 Experimental Setup

**Datasets.** For CIFAR-100 and *mini*ImageNet, we randomly select 60 classes as the base classes and the remaining 40 classes as the new classes. In each incremental learning session, we construct 5-way 5-shot tasks by randomly picking 5 classes and sampling 5 examples for each class. For CUB-200-2011 with 200 classes, we select 100 classes as the base classes and 100 classes as the new ones. We test 10-way 5-shot tasks on this dataset.

**Baselines.** We compare our method F2M with 8 methods: the Baseline proposed in Sec. 3, a joint-training method that uses all previously seen data including the base and the following few-shot

Table 1: Classification accuracy on CIFAR-100 for 5-way 5-shot incremental learning. * indicates our re-implementation.

| Method | sessions | | | | | | | | | The gap with cRT |
|--------|------|------|------|------|------|------|------|------|------|------|
| | 1 | 2 | 3 | 4 | 5 | 6 | 7 | 8 | 9 | |
| cRT [25]* | 65.18 | 63.89 | 60.20 | 57.23 | 53.71 | 50.39 | 48.77 | 47.29 | 45.28 | - |
| Joint-training* | 65.18 | 61.45 | 57.36 | 53.68 | 50.84 | 47.33 | 44.79 | 42.62 | 40.08 | -5.20 |
| Baseline | 65.18 | 61.67 | 58.61 | 55.11 | 51.86 | 49.43 | 47.60 | 45.64 | 43.83 | -1.45 |
| iCaRL [40]* | 66.52 | 57.26 | 54.27 | 50.62 | 47.33 | 44.99 | 43.14 | 41.16 | 39.49 | -5.79 |
| Rebalance [20]* | **66.66** | 61.42 | 57.29 | 53.02 | 48.85 | 45.68 | 43.06 | 40.56 | 38.35 | -6.93 |
| FSLL [34]* | 65.18 | 56.24 | 54.55 | 51.61 | 49.11 | 47.27 | 45.35 | 43.95 | 42.22 | -3.08 |
| iCaRL [40] | 64.10 | 53.28 | 41.69 | 34.13 | 27.93 | 25.06 | 20.41 | 15.48 | 13.73 | -31.55 |
| Rebalance [20] | 64.10 | 53.05 | 43.96 | 36.97 | 31.61 | 26.73 | 21.23 | 16.78 | 13.54 | -31.74 |
| TOPIC [47] | 64.10 | 55.88 | 47.07 | 45.16 | 40.11 | 36.38 | 33.96 | 31.55 | 29.37 | -15.91 |
| FSLL [34] | 64.10 | 55.85 | 51.71 | 48.59 | 45.34 | 43.25 | 41.52 | 39.81 | 38.16 | -7.12 |
| FSLL+SS [34] | 66.76 | 55.52 | 52.20 | 49.17 | 46.23 | 44.64 | 43.07 | 41.20 | 39.57 | -5.71 |
| **F2M** | 64.71 | **62.05** | **59.01** | **55.58** | **52.55** | **49.96** | **48.08** | **46.28** | **44.67** | **-0.61** |

Table 2: Classification accuracy on *mini*ImageNet for 5-way 5-shot incremental learning. * indicates our re-implementation.

| Method | sessions | | | | | | | | | The gap with cRT |
|--------|------|------|------|------|------|------|------|------|------|------|
| | 1 | 2 | 3 | 4 | 5 | 6 | 7 | 8 | 9 | |
| cRT [25]* | 67.30 | 64.15 | 60.59 | 57.32 | 54.22 | 51.43 | 48.92 | 46.78 | 44.85 | - |
| Joint-training* | 67.30 | 62.34 | 57.79 | 54.08 | 50.93 | 47.65 | 44.64 | 42.61 | 40.29 | -4.56 |
| Baseline | 67.30 | 63.18 | 59.62 | 56.33 | 53.28 | 50.50 | 47.96 | 45.85 | 43.88 | -0.97 |
| iCaRL [40]* | 67.35 | 59.91 | 55.64 | 52.60 | 49.43 | 46.73 | 44.13 | 42.17 | 40.29 | -4.56 |
| Rebalance [20]* | 67.91 | 63.11 | 58.75 | 54.83 | 50.68 | 47.11 | 43.88 | 41.19 | 38.72 | -6.13 |
| FSLL [34]* | 67.30 | 59.81 | 57.26 | 54.57 | 52.05 | 49.42 | 46.95 | 44.94 | 42.87 | -1.11 |
| iCaRL [40] | 61.31 | 46.32 | 42.94 | 37.63 | 30.49 | 24.00 | 20.89 | 18.80 | 17.21 | -27.64 |
| Rebalance [20] | 61.31 | 47.80 | 39.31 | 31.91 | 25.68 | 21.35 | 18.67 | 17.24 | 14.17 | -30.68 |
| TOPIC [47] | 61.31 | 50.09 | 45.17 | 41.16 | 37.48 | 35.52 | 32.19 | 29.46 | 24.42 | -20.43 |
| FSLL [34] | 66.48 | 61.75 | 58.16 | 54.16 | 51.10 | 48.53 | 46.54 | 44.20 | 42.28 | -2.57 |
| FSLL+SS [34] | **68.85** | 63.14 | 59.24 | 55.23 | 52.24 | 49.65 | 47.74 | 45.23 | 43.92 | -0.93 |
| IDLVQ-C [8] | 64.77 | 59.87 | 55.93 | 52.62 | 49.88 | 47.55 | 44.83 | 43.14 | 41.84 | -3.01 |
| **F2M** | 67.28 | **63.80** | **60.38** | **57.06** | **54.08** | **51.39** | **48.82** | **46.58** | **44.65** | **-0.20** |

tasks for training, the classifier re-training method (cRT) [25] for long-tailed classification trained with all encountered data, iCaRL [40], Rebalance [20], TOPIC [47], FSLL [34], and IDLVQ-C [8]. For a fair comparison, we re-implement cRT [25], iCaRL [40], Rebalance [20], FSLL [34], and the joint-training method and tune them to their best performance. We also provide the results reported in the original papers for comparison. The results of TOPIC [47] and IDLVQ-C [8] are copied from the original papers. Note that for IL, joint-training is naturally the upper bound of incremental learning algorithms, however, for IFL, joint-training is not a good approximation of the upper bound because data imbalance makes the model perform significantly poorer on new classes (long-tailed classes). To address the data imbalance issue, we re-implement the cRT method as the *approximate upper bound*.

**Experimental details.** The experiments are conducted with NVIDIA GPU RTX3090 on CUDA 11.0. We randomly split each dataset into multiple tasks (sessions). For each dataset (with a fixed split), we run each algorithm for 10 times and report the mean accuracy. We adopt ResNet18 [16] as the backbone network. For data augmentation, we use standard random crop and horizontal flip. In the base training stage, we select the last 4 or 8 convolution layers to inject noise, because these layers output higher-level feature representations. The flat region bound $b$ is set as 0.01. We set the number of times for noise sampling as $M = 2 \sim 4$, since a larger $M$ will increase the training time. In each incremental few-shot learning session, the total number of training epochs is 6, and the learning rate is 0.02. To verify the correctness of our implementation, we conduct experiments on incremental learning and compare our results to those reported on CIFAR-100 in Appendix A.3. More experiment details are provided in Appendix A.2.

Table 3: Classification accuracy on CUB-200-2011 for 10-way 5-shot incremental learning.* indicates our re-implementation.

| Method | sessions | | | | | | | | | | | The gap with cRT |
|---|---|---|---|---|---|---|---|---|---|---|---|---|
| | 1 | 2 | 3 | 4 | 5 | 6 | 7 | 8 | 9 | 10 | 11 | |
| cRT [25]* | 80.83 | 78.51 | 76.12 | 73.93 | 71.46 | 68.96 | 67.73 | 66.75 | 64.22 | 62.53 | 61.08 | - |
| Joint-training* | 80.83 | 77.57 | 74.11 | 70.75 | 68.52 | 65.97 | 64.58 | 62.22 | 60.18 | 58.49 | 56.78 | -4.30 |
| Baseline | 80.87 | 77.15 | 74.46 | 72.26 | 69.47 | 67.18 | 65.62 | 63.68 | 61.30 | 59.72 | 58.12 | -2.96 |
| iCaRL [40]* | 79.58 | 67.63 | 64.17 | 61.80 | 58.10 | 55.51 | 53.34 | 50.89 | 48.62 | 47.34 | 45.60 | -15.48 |
| Rebalance [20]* | 80.94 | 70.32 | 62.96 | 57.19 | 51.06 | 46.70 | 44.03 | 40.15 | 36.75 | 34.88 | 32.09 | -28.99 |
| FSLL [34]* | 80.83 | 77.38 | 72.37 | 71.84 | 67.51 | 65.30 | 63.75 | 61.16 | 59.05 | 58.03 | 55.82 | -5.26 |
| iCaRL [40] | 68.68 | 52.65 | 48.61 | 44.16 | 36.62 | 29.52 | 27.83 | 26.26 | 24.01 | 23.89 | 21.16 | -39.92 |
| Rebalance [20] | 68.68 | 57.12 | 44.21 | 28.78 | 26.71 | 25.66 | 24.62 | 21.52 | 20.12 | 20.06 | 19.87 | -41.21 |
| TOPIC [47] | 68.68 | 62.49 | 54.81 | 49.99 | 45.25 | 41.40 | 38.35 | 35.36 | 32.22 | 28.31 | 26.28 | -34.80 |
| FSLL [34] | 72.77 | 69.33 | 65.51 | 62.66 | 61.10 | 58.65 | 57.78 | 57.26 | 55.59 | 55.39 | 54.21 | -6.87 |
| FSLL+SS [34] | 75.63 | 71.81 | 68.16 | 64.32 | 62.61 | 60.10 | 58.82 | 58.70 | 56.45 | 56.41 | 55.82 | -5.26 |
| IDLVQ-C [8] | 77.37 | 74.72 | 70.28 | 67.13 | 65.34 | 63.52 | 62.10 | 61.54 | 59.04 | 58.68 | 57.81 | -3.27 |
| **F2M** | **81.07** | **78.16** | **75.57** | **72.89** | **70.86** | **68.17** | **67.01** | **65.26** | **63.36** | **61.76** | **60.26** | **-0.82** |

Table 4: Comparison of the flatness of the local minima found by the Baseline and our F2M.

| Method | Indicator $I$ | | Variance $\sigma^2$ | |
|---|---|---|---|---|
| | Training Set | Testing Set | Training Set | Testing Set |
| Baseline | 0.2993 | 0.4582 | 0.1451 | 0.2395 |
| **F2M** | **0.0506** | **0.0800** | **0.0296** | **0.0334** |

## 5.2 Comparison with the State-of-the-Art

**F2M outperforms the state-of-the-art methods**. The main results on CIFAR-100, miniImageNet and CUB-200-2011 are presented in Table 1, Table 2 and Table 3 respectively. Based on the experiment results, we have the following observations: **1)** The Baseline introduced in Sec. 3 outperforms the state-of-the-art approaches on all incremental sessions. **2)** As expected, cRT consistently outperforms the Baseline up to 1% to 3% by considering the data imbalance problem and applying proper techniques to tackle the long-tailed classification problem to improve performance. Hence, it is reasonable to use cRT as the approximate upper bound of IFL. **3)** Our F2M outperforms the state-of-the-art methods and the Baseline. Moreover, the performance of F2M is very close to the approximate upper bound, i.e., the gap with cRT is only 0.2% in the last session on *mini*ImageNet. The results show that even with strong constraints [20, 40, 34] and saved examplars of base classes [20, 40, 8], current methods cannot effectively address the catastrophic forgetting problem. In contrast, finding flat minima seems a promising approach to overcome this harsh problem.

## 5.3 Ablation Study and Analysis

**Analysis on the flatness of local minima.** Here, we verify that our method can find a more flat local minima than the Baseline. For a found local minima $\theta^\star$, we measure its flatness as follows. We sample the noise for 1000 times. For each time, we inject the sampled noise to $\theta^\star$ and calculate the loss $\mathcal{L}_i$. Then, we adopt the indicator $I = \frac{1}{1000}\sum_{i=1}^{1000}(\mathcal{L}_i - \mathcal{L}^*)^2$ and variance $\sigma^2 = \frac{1}{1000}\sum_{i=1}^{1000}(\mathcal{L}_i - \overline{\mathcal{L}})^2$ to measure the flatness. $\mathcal{L}^*$ denotes the loss of $\theta^\star$, and $\overline{\mathcal{L}}$ denotes the average loss of $\{\mathcal{L}_i\}_{i=1}^{1000}$. The values of the indicator and variance of F2M and the Baseline are presented in Table 4, which clearly demonstrate that our method can find a more flat local minima.

**Ablation study on the designs of our method.** Here, we study the effectiveness of each design of our method, including adding noise to the model parameters for finding $b$-flat local minima (FM) during the base training session, the prototype fixing term (PF) used in the base training objective (Eq. 4), parameter clamping (PC) during incremental learning, and prototype normalization (PN). We conduct an ablation study by removing each component in turn and report the experimental results in Table 5.

*Finding b-flat local minima.* Standard supervised training with SGD as the optimizer tends to converge to a sharp local minima. It leads to a significant drop in performance because the loss changes quickly

Table 5: Ablation study of our F2M on CIFAR-100. PD refers to the performance dropping rate.

| FM | PF | PC | PN | sessions | | | | | | | | | PD ↓ |
|---|---|---|---|---|---|---|---|---|---|---|---|---|---|
| | | | | 1 | 2 | 3 | 4 | 5 | 6 | 7 | 8 | 9 | |
| | | | | **65.18** | 60.83 | 53.13 | 43.57 | 23.75 | 10.76 | 08.26 | 07.24 | 06.45 | 58.73 |
| | | ✓ | | **65.18** | 59.48 | 56.77 | 52.99 | 50.09 | 47.80 | 45.92 | 44.20 | 42.55 | 22.63 |
| ✓ | ✓ | | ✓ | 64.71 | 59.54 | 53.03 | 45.09 | 41.68 | 39.04 | 38.64 | 37.19 | 36.01 | 28.70 |
| ✓ | | ✓ | ✓ | 64.55 | 61.27 | 58.33 | 54.82 | 51.60 | 49.22 | 47.48 | 45.78 | 44.08 | 20.47 |
| ✓ | ✓ | ✓ | | 64.71 | 61.75 | 58.80 | 55.33 | 52.27 | 49.75 | 47.72 | 46.01 | 44.43 | 20.28 |
| ✓ | ✓ | ✓ | ✓ | 64.71 | **61.99** | **58.99** | **55.58** | **52.55** | **49.96** | **48.08** | **46.28** | **44.67** | **20.04** |

Table 6: Study of the flat region bound $b$ for 5-way 5-shot incremental learning on CIFAR-100. The top 3 results in each row are in boldface.

| Session | The hyperparameter $b$ | | | | | |
|---|---|---|---|---|---|---|
| | 0.0025 | 0.005 | 0.01 | 0.02 | 0.04 | 0.08 |
| Session 1 (60 bases classes) | **64.85** | 64.67 | **64.81** | **64.71** | 63.30 | 62.25 |
| Session 9 (All 100 classes) | 44.16 | **44.54** | **44.58** | **44.67** | 43.75 | 43.04 |
| Session 9 (60 base classes) | **59.58** | **59.69** | **59.73** | 59.44 | 58.38 | 57.21 |
| Session 9 (40 new classes) | 21.03 | **21.81** | **21.86** | **22.52** | 21.80 | 21.77 |

in the neighborhood of the sharp local minima. As shown in Table 5, even with parameter clamping during incremental learning, the performance still drops significantly. In contrast, restricting the parameters in a small flat region can mitigate the forgetting problem.

*Prototype fixing.* Without fixing the prototypes after injecting noise to selected layers during the process of finding local minima, i.e. removing the second term of Eq. 4, it is still possible to tune the model within the flat region to well separate base classes. However, the saved prototypes of base classes will become less accurate because the embeddings of the base samples suffer from semantic drift [54]. As shown in Table 5, it results in a performance drop of nearly $0.6\%$.

*Parameter clamping.* Parameter clamping restricts the model parameters to the $b$-flat region after incremental few-shot learning. Outside the $b$-flat region, the performance drops quickly. It can be seen from Table 5 that removing parameter clamping leads to a significant drop in performance.

*Prototype normalization.* As mentioned in Sec. 4.2, we normalize the class prototypes to calibrate the estimates of the class mean classifier. The results in Table 5 show the effectiveness of normalization, which helps to further improve the performance.

**Study of the flat region bound** $b$**.** We study the effect of the flat region bound $b$ for 5-way 5-shot incremental learning on CIFAR-100. We report the test accuracy in session 1 (base session) and session 9 (last session) w.r.t. different $b$ in Table 6. It can be seen that the best results are achieved for $b \in [0.005, 0.02]$. A larger $b$ (e.g., 0.04 or 0.08) leads to a significant performance drop on base classes, even for those in session 1, indicating that there may not exist a large flat region around a good local minima. Meanwhile, a smaller $b$ (e.g., 0.0025) results in a performance decline on new classes, due to the overly small capacity of the flat region. This illustrates the trade-off effect of $b$.

## 6 Conclusion

We have proposed a novel approach to overcome catastrophic forgetting in incremental few-shot learning by finding flat local minima of the objective function in the base training stage and then fine-tuning the model within the flat region on new tasks. Extensive experiments on benchmark datasets show that our model can effectively mitigate catastrophic forgetting and adapt to new classes. A limitation of our method is that it may not be suitable for medium- or high-shot tasks, since the flat region is relatively small, which limits the model capacity. However, it is still possible to adapt our core idea for incremental learning. For example, one can search for a less flat but wider local minima region in the base training stage and tune the model within this region during incremental learning sessions, where previous techniques such as elastic weight consolidation (EWC) [28] can be used to constraint the model parameters. This could be an interesting direction for future research.

## Acknowledgments and Disclosure of Funding

We would like to thank the anonymous reviewers for their insightful and helpful comments. This research was supported by the grant of DaSAIL project P0030935 funded by PolyU/UGC.

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
