# A Appendix

## A.1 Proof of Theorem 4.1

**Lemma A.1.** *By Assumption 4.1 and 4.2, we have*

$$\mathbb{E}_{z_k,\epsilon_j}[R(\theta_{k+1})] - R(\theta_k) \leq -\alpha_k \frac{2M - \alpha_k L(m_2 + M)}{2M} \|\nabla R(\theta_k)\|_2^2 + \frac{\alpha_k^2 L m_1}{2M}. \tag{12}$$

*Proof.* By Assumption 4.1, an important consequence is that for all $\{\theta, \theta'\} \subset \mathbb{R}^d$, it satisfies that

$$R(\theta) \leq R(\theta') + \nabla R(\theta')^T(\theta - \theta') + \frac{1}{2}L\|\theta - \theta'\|_2^2. \tag{13}$$

Taken together, the above inequality and the parameter update equation (Eq. 6), it yields

$$R(\theta_{k+1}) - R(\theta_k) \leq \nabla R(\theta_k)^T(\theta_{k+1} - \theta_k) + \frac{1}{2}L\|\theta_{k+1} - \theta_k\|_2^2 \leq -\alpha_k \nabla R(\theta_k)^T \overline{g} + \frac{\alpha_k^2 L}{2}\|\overline{g}\|_2^2, \tag{14}$$

where $\overline{g} = \frac{1}{M}\sum_{j=1}^{M} g(z_k; \phi_k + \epsilon_j, \psi_k)$. Taking expectation on both sides of Eq. 14, it yields

$$\mathbb{E}_{z_k,\epsilon_j}[R(\theta_{k+1})] - R(\theta_k) \leq -\alpha_k \nabla R(\theta_k)^T \mathbb{E}_{z_k,\epsilon_j}[\overline{g}] + \frac{\alpha_k^2 L}{2}\mathbb{E}_{z_k,\epsilon_j}[\|\overline{g}\|_2^2]. \tag{15}$$

$\mathbb{E}_{z_k,\epsilon_j}[\cdot]$ denotes the expectation w.r.t. the joint distribution of random variables $z_k$ and $\epsilon_j$ given $\theta_k$. Note that $\theta_{k+1}$ (not $\theta_k$) depends on $z_k$ and $\epsilon_j$. Under Condition 2 of Assumption 4.2, the expectation of $\overline{g}$ satisfies that

$$\mathbb{E}_{z_k,\epsilon_j}[\overline{g}] = \frac{1}{M}\sum_{j=1}^{M}\mathbb{E}_{z_k,\epsilon_j}[g(z_k; \phi_k + \epsilon_j, \psi_k)] = \nabla R(\theta_k). \tag{16}$$

Assume that we sample the noise vector $\epsilon_j$ from $P(\epsilon)$ *without replacement*. Under Condition 3 of Assumption 4.2, we have (see [1, p. 183])

$$\mathbb{V}_{z_k,\epsilon_j}[\overline{g}] \leq \frac{\mathbb{V}_{z_k,\epsilon_j}[g(z_k; \phi_k + \epsilon_j, \psi_k)]}{M} \leq \frac{m_1}{M} + \frac{m_2}{M}\|\nabla R(\theta_k)\|_2^2. \tag{17}$$

Taken together, Eq. 16 and Eq. 17, one obtains

$$\mathbb{E}_{z_k,\epsilon_j}[\|\overline{g}\|_2^2] = \mathbb{V}_{z_k,\epsilon_j}[\overline{g}] + \|\mathbb{E}_{z_k,\epsilon_j}[\overline{g}]\|_2^2 \leq \frac{m_1}{M} + \frac{m_2 + M}{M}\|\nabla R(\theta_k)\|_2^2. \tag{18}$$

Therefore, by Eq. 15, 16 and 18, it yields

$$\mathbb{E}_{z_k,\epsilon_j}[R(\theta_{k+1})] - R(\theta_k) \leq -\alpha_k \frac{2M - \alpha_k L(m_2 + M)}{2M}\|\nabla R(\theta_k)\|_2^2 + \frac{\alpha_k^2 L m_1}{2M}. \tag{19}$$

$\square$

**Lemma A.2.** *By Assumption 4.1, 4.2 and 4.3, we have*

$$\liminf_{k\to\infty}\mathbb{E}[\|\nabla R(\theta_k)\|_2^2] = 0. \tag{20}$$

*Proof.* The first condition in Assumption 4.3 ensures that $\lim_{k\to\infty}\alpha_k = 0$. Without loss of generality, we assume that for any $k \in \mathbb{N}$, $\alpha_k L(m_2 + M) \leq M$. Denote by $\mathbb{E}[\cdot]$ the *total expectation* w.r.t. all involved random variables. For example, $\theta_k$ is determined by the set of random variables $\{z_0, z_1, ..., z_{k-1}, \epsilon_0, \epsilon_1, ..., \epsilon_{k-1}\}$, and therefore the *total expectation* of $R(\theta_k)$ is given by

$$\mathbb{E}[R(\theta_k)] = \mathbb{E}_{z_0,\epsilon_0}\mathbb{E}_{z_1,\epsilon_1}...\mathbb{E}_{z_{k-1},\epsilon_{k-1}}[R(\theta_k)]. \tag{21}$$

Taking total expectation on both sides of Eq.12, we have

$$\mathbb{E}[R(\theta_k + 1)] - \mathbb{E}[R(\theta_k)] \leq -\frac{\alpha_k}{2}\mathbb{E}[\|\nabla R(\theta_k)\|_2^2] + \frac{\alpha_k^2 L m_1}{2M}. \tag{22}$$

For $k = 0, 1, 2, ..., K$, summing both sides of this inequality yields

$$R^\star - \mathbb{E}[R(\theta_1)] \leq \mathbb{E}[R(\theta_{K+1})] - \mathbb{E}[R(\theta_0)] \leq -\frac{1}{2}\sum_{k=0}^{K}\alpha_k\mathbb{E}[\|\nabla R(\theta_k)\|_2^2] + \frac{Lm_1}{2M}\sum_{k=0}^{K}\alpha_k^2, \quad (23)$$

where $R^\star$ is the lower bound in Condition 1 of Assumption 4.2. Rearranging the term gives

$$\sum_{k=0}^{K}\alpha_k\mathbb{E}[\|\nabla R(\theta_k)\|_2^2] \leq 2(\mathbb{E}[R(\theta_1)] - R^\star) + \frac{Lm_1}{M}\sum_{k=0}^{K}\alpha_k^2. \quad (24)$$

By the second condition of Assumption 4.3, we have

$$\lim_{K\to\infty}\mathbb{E}[\sum_{k=0}^{K}\alpha_k\|\nabla R(\theta_k)\|_2^2] \leq 2(\mathbb{E}[R(\theta_0)] - R^\star) + \lim_{K\to\infty}\frac{Lm_1}{M}\sum_{k=0}^{K}\alpha_k^2 < \infty. \quad (25)$$

Dividing both sides of Eq. 25 by $\sum_{k=1}^{K}\alpha_k$ and by the first condition of Assumption 4.3, we have

$$\lim_{K\to\infty}\mathbb{E}[\frac{\sum_{k=1}^{K}\alpha_k\|\nabla R(\theta_k)\|_2^2}{\sum_{k=1}^{K}\alpha_k}] = 0. \quad (26)$$

The left-hand term of this equation is the weighed average of $\|\nabla R(\theta_k)\|_2^2$, and $\{\alpha_k\}$ are the weights. Hence, a direct consequence of this equation is that $\|\nabla R(\theta_k)\|_2^2$ cannot asymptotically stay far from zero, i.e.

$$\liminf_{k\to\infty}\mathbb{E}[\|\nabla R(\theta_k)\|_2^2] = 0. \quad (27)$$

$$\square$$

We now prove Theorem 4.1, which is a stronger consequence than Lemma A.2.

**Theorem 4.1.** *Under assumptions 4.1, 4.2 and 4.3, we further assume that the risk function $R$ is twice differentiable, and that $\|\nabla R(\theta)\|_2^2$ is $L_2$-smooth with constant $L_2 > 0$, then we have*

$$\lim_{k\to\infty}\mathbb{E}[\|\nabla R(\theta_k)\|_2^2] = 0. \quad (28)$$

*Proof.* Define $F(\theta) \coloneqq \|R(\theta)\|_2^2$, then we have

$$\mathbb{E}_{z_k,\epsilon_j}[F(\theta_{k+1})] - F(\theta_k) \leq \nabla F(\theta_k)^T\mathbb{E}_{z_k,\epsilon_j}[(\theta_{k+1} - \theta_k)] + \frac{1}{2}L_2\mathbb{E}_{z_k,\epsilon_j}[\|\theta_{k+1} - \theta_k\|_2^2]$$

$$\leq -\alpha_k\nabla F(\theta_k)^T\mathbb{E}_{z_k,\epsilon_j}[\overline{g}] + \frac{\alpha_k^2L_2}{2}\mathbb{E}_{z_k,\epsilon_j}[\|\overline{g}\|_2^2]$$

$$\leq -2\alpha_k\nabla R(\theta_k)^T\nabla^2 R(\theta_k)^T\mathbb{E}_{z_k,\epsilon_j}[\overline{g}] + \frac{\alpha_k^2L_2}{2}\mathbb{E}_{z_k,\epsilon_j}[\|\overline{g}\|_2^2]$$

$$\leq 2\alpha_k\|\nabla R(\theta_k)\|_2^2\|\nabla^2 R(\theta_k)\|_2\|\mathbb{E}_{z_k,\epsilon_j}[\overline{g}]\|_2 + \frac{\alpha_k^2L_2}{2}\mathbb{E}_{z_k,\epsilon_j}[\|\overline{g}\|_2^2]$$

$$\leq 2\alpha_kL\|\nabla R(\theta_k)\|_2^2 + \frac{\alpha_k^2L_2}{2}(\frac{m_1}{M} + \frac{m_2+M}{M}\|\nabla R(\theta_k)\|_2^2). \quad (29)$$

Taking total expectation of both sides of Eq. 29 yields

$$\mathbb{E}[F(\theta_{k+1})] - \mathbb{E}[F(\theta_k)] \leq 2\alpha_kL\mathbb{E}[\|\nabla R(\theta_k)\|_2^2] + \frac{\alpha_k^2L_2}{2}(\frac{m_1}{M} + \frac{m_2+M}{M}\mathbb{E}[\|\nabla R(\theta_k)\|_2^2]). \quad (30)$$

Eq. 25 implies that $2\alpha_kL\mathbb{E}[\|\nabla R(\theta_k)\|_2^2]$ is the term of a convergent sum. Besides, $\frac{\alpha_k^2L_2}{2}(\frac{m_1}{M} + \frac{m_2+M}{M}\mathbb{E}[\|\nabla R(\theta_k)\|_2^2])$ is also the term of a convergent sum, because $\sum_{k=1}^{\infty}\alpha_k^2$ converges. Hence, the bound (Eq. 30) is also the term of a convergent sum. Now, let us define

$$A_K^+ = \sum_{k=0}^{K-1}\max(0, \mathbb{E}[F(\theta_{k+1})] - \mathbb{E}[F(\theta_k)]), \quad (31)$$

$$\text{and} \quad A_K^- = \sum_{k=0}^{K-1}\max(0, \mathbb{E}[F(\theta_k)] - \mathbb{E}[F(\theta_{k+1})]). \quad (32)$$

Because the bound of $\mathbb{E}[F(\theta_{k+1})] - \mathbb{E}[F(\theta_k)]$ is positive and is the term a of convergent sum, and the sequence $A_K^+$ is upper bounded by the sum of the bound of $\mathbb{E}[F(\theta_{k+1})] - \mathbb{E}[F(\theta_k)]$, $A_K^+$ converges. Similarly, $A_K^-$ also converges. Since for any $K \in \mathbb{N}$, $F(\theta_K) = F(\theta_0) + A_K^+ - A_K^-$, we can obtain that $F(\theta_k)$ converges. By Lemma A.2 and the fact that $F(\theta_k)$ converges, we have

$$\lim_{k \to \infty} \mathbb{E}[\|R(\theta_k)\|_2^2] = 0. \tag{33}$$

$\square$

### A.2 More Experimental Details

In CIFAR-100, each class contains 500 images for training and 100 images for test, with each image of size 32×32. In *mini*ImageNet, each class contains 500 training images and 100 test images of size 84×84. CUB-200-2011 contains 5994 training images and 5794 test images in total with varying number of images for each class, and we resize and crop each image to be of size $224 \times 224$.

Since there lacks a unified standard in storing/saving exemplars for incremental few-shot learning, we choose the setting that we consider most reasonable and practical. In real-world applications, normally there exists a large number of base classes with sufficient training data (e.g., the base dataset is ImageNet-1K [5]), whereas the number of unseen novel classes that lack training data is relatively small. Therefore, for computational efficiency and efficient use of storage, it is desirable NOT saving any exemplars for base classes but store some exemplars for new classes. In our experiments, we do not store any exemplar for base classes, but save 5 exemplars for each new class. This will hardly cost any storage space or slow down computation considerably due to the small number of new classes.

*To ensure a fair comparison*, for ICaRL [4] and Rebalance [2], we store 2 exemplars per class (for both base classes and new classes). As a result, in each session, they store more examplars than our method. For our re-implementation of FSLL [3], we store the same number of exemplars for each new class as in our method. For other approaches, since the code is not available or the method is too complex to re-implement, we directly use the results reported in their paper, which are substantially lower than the Baseline.

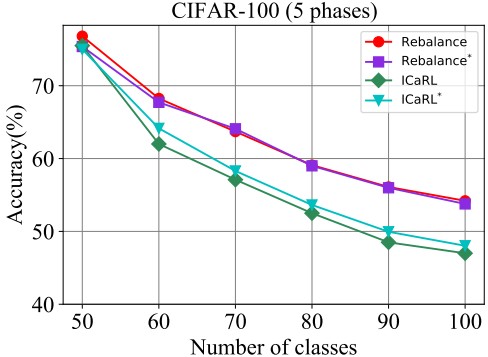

Figure 3: Our re-implementation results of Rebalance and ICaRL are very close to those reported in [2]. * indicates our re-implementation.

Table 7: The average norm of the class prototypes of new classes is significantly smaller than that of old classes. The experiment is conducted on CIFAR-100 with 60 base classes and 40 new classes.

|  | Mean | Standard Deviation |
|---|---|---|
| Base classes | 7.97 | 0.63 |
| New classes | 7.48 | 0.71 |

### A.3 Additional Experiment Results

In this section, we conduct experiments to verify the correctness of our re-implementation of state-of-the-art methods including ICaRL* [4], Rebalance* [2] and FSLL* [3]. Note that we re-implement ICaRL and Rebalance because they (and the released codes) are designed for incremental learning, not for incremental few-shot learning. We re-implement FSLL because the code is not provided. In addition, we present empirical evidence on the difference in the norm of the class prototypes between new classes and base classes, which motivates the design of prototype normalization.

**Correctness of our implementation.** To verify the correctness of our implementation of ICaRL* [4] and Rebalance* [2], we conduct experiments on CIFAR-100 for *incremental learning*. We adopt 32-layer ResNet as backbone and store 20 exemplars per class as in Rebalance [2]. The comparative

Table 8: Our re-implementation results of FSLL are very close to those reported in [3] on CIFAR-100 for 5-way 5-shot incremental learning. * indicates our re-implementation. The results are obtained without saving any exemplars.

| Method | sessions | | | | | | | | |
|--------|------|------|------|------|------|------|------|------|------|
| | 1 | 2 | 3 | 4 | 5 | 6 | 7 | 8 | 9 |
| FSLL [3]* | 65.18 | 56.37 | 52.59 | 48.39 | 47.46 | 43.44 | 41.37 | 40.17 | 38.56 |
| FSLL [3] | 64.10 | 55.85 | 51.71 | 48.59 | 45.34 | 43.25 | 41.52 | 39.81 | 38.16 |

results are presented in Fig 3. It can be seen that our re-implementation results of ICaRL and Rebalance are very close to those reported in [2].

To verify the correctness of our implementation of FSLL [3], we compare the results of our implementation and those reported in [3] in Table 8. It can be seen that our implementation achieves similar and slightly higher results than those reported in the original paper [3]. Here, the experiments are conducted following the settings in [3] without saving any exemplars for new classes.

**Norm of class prototype.** In our experiments, we observe that after training on base classes with balanced data, the norms of the class prototypes of base classes tend to be similar. However, after fine-tuning with very few data on unseen new classes, the norms of the new class prototypes are noticeably smaller than those of the base classes. In Table 7, we show the average norms of the prototypes of base classes and new classes after incremental few-shot learning on CIFAR-100, where we randomly select 60 classes as base classes and the remaining 40 classes as new classes.

Table 9: Classification accuracy for 5-way 5-shot incremental learning with the first class split on CIFAR-100.

| Method | sessions | | | | | The gap with cRT |
|--------|----------|---|---|---|---|----------|
| | 1 (Animals) | 2 (Vehicles2) | 3 (Flowers) | 4 (Food Containers) | 5 (Household Furniture) | |
| Baseline | 63.07 | 56.32 | 51.40 | 46.85 | 43.55 | -19.52 |
| ICaRL | **63.30** | 55.10 | 49.12 | 44.46 | 40.95 | -22.35 |
| Rebalance | 63.03 | 52.06 | 45.87 | 39.35 | 35.24 | -27.29 |
| FSLL | 63.07 | 50.72 | 44.53 | 40.73 | 38.00 | -25.07 |
| **F2M** | 62.53 | **56.63** | **51.87** | **47.54** | **44.10** | **-18.43** |

Table 10: Classification accuracy for 5-way 5-shot incremental learning with the second class split on CIFAR-100.

| Method | sessions | | | | | The gap with cRT |
|--------|----------|---|---|---|---|----------|
| | 1 (Animals+Furniture) | 2 (People) | 3 (Vehicles2) | 4 (Flowers) | 5 (Food Containers) | |
| Baseline | 63.07 | 54.30 | 50.16 | 46.19 | 43.16 | -19.91 |
| ICaRL | 62.57 | 51.67 | 47.51 | 42.98 | 39.63 | -22.94 |
| Rebalance | **63.50** | 49.62 | 44.67 | 39.68 | 35.64 | -27.86 |
| FSLL | 63.07 | 49.45 | 46.30 | 41.94 | 39.33 | -23.74 |
| **F2M** | 62.87 | **54.82** | **50.88** | **46.88** | **43.83** | **-19.04** |

**Results on CIFAR-100 with different class splits.** To analyze how difference in patterns of the base and new classes influence our proposed method F2M, we split the classes according to the superclasses and provide results on two different class splits. All the 100 classes of CIFAR-100 are grouped into 20 superclasses, and each superclass contains 5 classes. For the first class split, the base classes consist of aquatic mammals, fish, insects, reptiles, small mammals, and large carnivores (30 classes in total). The few-shot novel classes consist of household furniture, vehicles2, flowers, and food containers (20 classes in total). For the second class split, the base classes consist of aquatic mammals, fish, insects, reptiles, household furniture, and small mammals (30 classes in total). The few-shot novel classes consist of people, vehicles2, flowers, and food containers (20 classes in total). The experimental results with the two different class splits are presented in Table 9 and Table 10 respectively. The results show that even with a large difference between the base classes and novel classes, our F2M still consistently outperforms other methods, indicating its robustness and effectiveness.

**Error bars of the main results.** The experimental results reported in Section 5 are the average of 10 runs. For each run, we randomly selected 5 samples for each class (for 5-shot tasks). Here, in Table 11, Table 12 and Table 13, we report the means and 95% confidence intervals of our method F2M, the Baseline, and the methods that we re-implemented. The confidence intervals indicate that our method F2M achieves steady improvement over state-of-the-art methods.

Table 11: Classification accuracy on CIFAR-100 for 5-way 5-shot incremental learning with 95% confidence intervals. * indicates our re-implementation.

| Method | sessions | | | | | | | | |
|---|---|---|---|---|---|---|---|---|---|
| | 1 | 2 | 3 | 4 | 5 | 6 | 7 | 8 | 9 |
| Baseline | 65.18 | 61.67 ± 0.18 | 58.61 ± 0.25 | 55.11 ± 0.19 | 51.86 ± 0.22 | 49.43 ± 0.28 | 47.60 ± 0.25 | 45.64 ± 0.29 | 43.83 ± 0.22 |
| iCaRL [4]* | 66.52 | 57.26 ± 0.17 | 54.27 ± 0.25 | 50.62 ± 0.29 | 47.33 ± 0.27 | 44.99 ± 0.26 | 43.14 ± 0.23 | 41.16 ± 0.30 | 39.49 ± 0.30 |
| Rebalance [2]* | **66.66** | 61.42 ± 0.25 | 57.29 ± 0.17 | 53.02 ± 0.20 | 48.85 ± 0.21 | 45.68 ± 0.30 | 43.06 ± 0.27 | 40.56 ± 0.38 | 38.35 ± 0.48 |
| FSLL [3]* | 65.18 | 56.24 ± 0.35 | 54.55 ± 0.28 | 51.61 ± 0.36 | 49.11 ± 0.40 | 47.27 ± 0.29 | 45.35 ± 0.32 | 43.95 ± 0.28 | 42.22 ± 0.49 |
| **F2M** | 64.71 | **62.05 ± 0.19** | **59.01 ± 0.22** | **55.58 ± 0.21** | **52.55 ± 0.25** | **49.96 ± 0.21** | **48.08 ± 0.24** | **46.28 ± 0.24** | **44.67 ± 0.19** |

Table 12: Classification accuracy on miniImageNet for 5-way 5-shot incremental learning with 95% confidence intervals. * indicates our re-implementation.

| Method | sessions | | | | | | | | |
|---|---|---|---|---|---|---|---|---|---|
| | 1 | 2 | 3 | 4 | 5 | 6 | 7 | 8 | 9 |
| Baseline | 67.30 | 63.18 ± 0.00 | 59.62 ± 0.12 | 56.33 ± 0.18 | 53.28 ± 0.27 | 50.50 ± 0.28 | 47.96 ± 0.30 | 45.85 ± 0.32 | 43.88 ± 0.27 |
| iCaRL [4]* | 67.35 | 59.91 ± 0.15 | 55.64 ± 0.20 | 52.60 ± 0.30 | 49.43 ± 0.32 | 46.73 ± 0.28 | 44.13 ± 0.33 | 42.17 ± 0.33 | 40.29 ± 0.31 |
| Rebalance [2]* | **67.91** | 63.11 ± 0.19 | 58.75 ± 0.29 | 54.83 ± 0.37 | 50.68 ± 0.38 | 47.11 ± 0.36 | 43.88 ± 0.33 | 41.19 ± 0.38 | 38.72 ± 0.39 |
| FSLL [3]* | 67.30 | 59.81 ± 0.42 | 57.26 ± 0.55 | 54.57 ± 0.58 | 52.05 ± 0.49 | 49.42 ± 0.37 | 46.95 ± 0.36 | 44.94 ± 0.20 | 42.87 ± 0.25 |
| **F2M** | 67.28 | **63.80 ± 0.10** | **60.38 ± 0.19** | **57.06 ± 0.29** | **54.08 ± 0.28** | **51.39 ± 0.32** | **48.82 ± 0.32** | **46.58 ± 0.33** | **44.65 ± 0.29** |

Table 13: Classification accuracy on CUB-200-2011 for 10-way 5-shot incremental learning with 95% confidence intervals. * indicates our re-implementation.

| Method | sessions | | | | | | | | | | |
|---|---|---|---|---|---|---|---|---|---|---|---|
| | 1 | 2 | 3 | 4 | 5 | 6 | 7 | 8 | 9 | 10 | 11 |
| Baseline | 80.87 | 77.15 ± 0.18 | 74.46 ± 0.22 | 72.26 ± 0.26 | 69.47 ± 0.35 | 67.18 ± 0.27 | 65.62 ± 0.38 | 63.68 ± 0.25 | 61.30 ± 0.22 | 59.72 ± 0.27 | 58.12 ± 0.27 |
| iCaRL [4]* | 79.58 | 67.63 ± 0.25 | 64.17 ± 0.30 | 61.80 ± 0.35 | 58.10 ± 0.33 | 55.51 ± 0.38 | 53.34 ± 0.32 | 50.89 ± 0.25 | 48.62 ± 0.29 | 47.34 ± 0.33 | 45.60 ± 0.31 |
| Rebalance [2]* | 80.94 | 70.32 ± 0.28 | 62.96 ± 0.31 | 57.19 ± 0.30 | 51.06 ± 0.37 | 46.70 ± 0.29 | 44.03 ± 0.40 | 40.15 ± 0.27 | 36.75 ± 0.32 | 34.88 ± 0.35 | 32.09 ± 0.39 |
| FSLL [3]* | 80.83 | 77.38 ± 0.30 | 72.37 ± 0.25 | 71.84 ± 0.45 | 67.51 ± 0.42 | 65.30 ± 0.50 | 63.75 ± 0.39 | 61.16 ± 0.28 | 59.05 ± 0.37 | 58.03 ± 0.35 | 55.82 ± 0.33 |
| **F2M** | **81.07** | **78.16 ± 0.14** | **75.57 ± 0.24** | **72.89 ± 0.32** | **70.86 ± 0.25** | **68.17 ± 0.39** | **67.01 ± 0.32** | **65.26 ± 0.26** | **63.36 ± 0.24** | **61.76 ± 0.27** | **60.26 ± 0.28** |

**Results with the same class splits as in TOPIC [6].** The experimental results of our F2M and some other methods (our re-implementations) presented in Table 1, Table 2, and Table 3 are on random class splits with random seed 1997. Here, we conduct experiments using the same class split as in TOPIC [6]. The experimental results on CIFAR-100, *mini*ImageNet, and CUB-200-2011 are presented in Table 14, Table 15, and Table 16 respectively. The results show that the Baseline and our F2M still consistently outperform other methods. Note that on CUB-200-2011, joint-training outperforms the Baseline and our F2M. The reasons may include: 1) The data imbalance issue is not very significant since the average number of images per class of this dataset is relatively small

(about 30); and 2) During the base training stage, we use a smaller learning rate (e.g., 0.001) for the embedding network (pretrained on ImageNet) and a higher learning rate (e.g., 0.01) for the classifier.

Table 14: Classification accuracy on CIFAR-100 for 5-way 5-shot incremental learning with the same class split as in TOPIC [6]. * indicates our re-implementation.

| Method | sessions | | | | | | | | | The gap with cRT |
|---|---|---|---|---|---|---|---|---|---|---|
| | 1 | 2 | 3 | 4 | 5 | 6 | 7 | 8 | 9 | |
| cRT [8]* | 72.28 | 69.58 | 65.16 | 61.41 | 58.83 | 55.87 | 53.28 | 51.38 | 49.51 | - |
| Joint-training* | 72.28 | 68.40 | 63.31 | 59.16 | 55.73 | 52.81 | 49.01 | 46.74 | 44.34 | -5.17 |
| Baseline | 72.28 | 68.01 | 64.18 | 60.56 | 57.44 | 54.69 | 52.98 | 50.80 | 48.70 | -0.81 |
| iCaRL [4]* | 72.05 | 65.35 | 61.55 | 57.83 | 54.61 | 51.74 | 49.71 | 47.49 | 45.03 | -4.48 |
| Rebalance [2]* | **74.45** | 67.74 | 62.72 | 57.14 | 52.78 | 48.62 | 45.56 | 42.43 | 39.22 | -10.29 |
| FSLL [3]* | 72.28 | 63.84 | 59.64 | 55.49 | 53.21 | 51.77 | 50.93 | 48.94 | 46.96 | -2.55 |
| iCaRL [4] | 64.10 | 53.28 | 41.69 | 34.13 | 27.93 | 25.06 | 20.41 | 15.48 | 13.73 | -35.78 |
| Rebalance [2] | 64.10 | 53.05 | 43.96 | 36.97 | 31.61 | 26.73 | 21.23 | 16.78 | 13.54 | -35.97 |
| TOPIC [6] | 64.10 | 55.88 | 47.07 | 45.16 | 40.11 | 36.38 | 33.96 | 31.55 | 29.37 | -20.14 |
| FSLL [3] | 64.10 | 55.85 | 51.71 | 48.59 | 45.34 | 43.25 | 41.52 | 39.81 | 38.16 | -11.35 |
| FSLL+SS [3] | 66.76 | 55.52 | 52.20 | 49.17 | 46.23 | 44.64 | 43.07 | 41.20 | 39.57 | -9.94 |
| **F2M** | 71.45 | **68.10** | **64.43** | **60.80** | **57.76** | **55.26** | **53.53** | **51.57** | **49.35** | **-0.16** |

Table 15: Classification accuracy on *mini*ImageNet for 5-way 5-shot incremental learning with the same class split as in TOPIC [6]. * indicates our re-implementation.

| Method | sessions | | | | | | | | | The gap with cRT |
|---|---|---|---|---|---|---|---|---|---|---|
| | 1 | 2 | 3 | 4 | 5 | 6 | 7 | 8 | 9 | |
| cRT [8]* | 72.08 | 68.15 | 63.06 | 61.12 | 56.57 | 54.47 | 51.81 | 49.86 | 48.31 | - |
| Joint-training* | 72.08 | 67.31 | 62.04 | 58.51 | 54.41 | 51.53 | 48.70 | 45.49 | 43.88 | -4.43 |
| Baseline | 72.08 | 66.29 | 61.99 | 58.71 | 55.73 | 53.04 | 50.40 | 48.59 | 47.31 | -1.0 |
| iCaRL [4]* | 71.77 | 61.85 | 58.12 | 54.60 | 51.49 | 48.47 | 45.90 | 44.19 | 42.71 | -5.6 |
| Rebalance [2]* | **72.30** | 66.37 | 61.00 | 56.93 | 53.31 | 49.93 | 46.47 | 44.13 | 42.19 | -6.12 |
| FSLL [3]* | 72.08 | 59.04 | 53.75 | 51.17 | 49.11 | 47.21 | 45.35 | 44.06 | 43.65 | -4.66 |
| iCaRL [4] | 61.31 | 46.32 | 42.94 | 37.63 | 30.49 | 24.00 | 20.89 | 18.80 | 17.21 | -31.10 |
| Rebalance [2] | 61.31 | 47.80 | 39.31 | 31.91 | 25.68 | 21.35 | 18.67 | 17.24 | 14.17 | -34.14 |
| TOPIC [6] | 61.31 | 50.09 | 45.17 | 41.16 | 37.48 | 35.52 | 32.19 | 29.46 | 24.42 | -23.89 |
| FSLL [3] | 66.48 | 61.75 | 58.16 | 54.16 | 51.10 | 48.53 | 46.54 | 44.20 | 42.28 | -6.03 |
| FSLL+SS [3] | 68.85 | 63.14 | 59.24 | 55.23 | 52.24 | 49.65 | 47.74 | 45.23 | 43.92 | -4.39 |
| **F2M** | 72.05 | **67.47** | **63.16** | **59.70** | **56.71** | **53.77** | **51.11** | **49.21** | **47.84** | **-0.43** |

Table 16: Classification accuracy on CUB-200-2011 for 10-way 5-shot incremental learning with the same class split as in TOPIC [6]. * indicates our re-implementation.

| Method | sessions | | | | | | | | | | | The gap with cRT |
|---|---|---|---|---|---|---|---|---|---|---|---|---|
| | 1 | 2 | 3 | 4 | 5 | 6 | 7 | 8 | 9 | 10 | 11 | |
| cRT [8]* | 77.16 | 74.41 | 71.31 | 68.08 | 65.57 | 63.08 | 62.44 | 61.29 | 60.12 | 59.85 | 59.30 | - |
| Joint-training* | 77.16 | 74.39 | 69.83 | 67.17 | 64.72 | 62.25 | 59.77 | 59.05 | 57.99 | 57.81 | 56.82 | -2.48 |
| Baseline | 77.16 | **74.00** | 70.21 | 66.07 | 63.90 | 61.35 | 60.01 | 58.66 | 56.33 | 56.12 | 55.07 | -4.23 |
| iCaRL [4]* | 75.95 | 60.90 | 57.65 | 54.51 | 50.83 | 48.21 | 46.95 | 45.74 | 43.21 | 43.01 | 41.27 | -18.03 |
| Rebalance [2]* | **77.44** | 58.10 | 50.15 | 44.80 | 39.12 | 34.44 | 31.73 | 29.75 | 27.56 | 26.93 | 25.30 | -34.00 |
| FSLL [3]* | 77.16 | 71.85 | 66.53 | 59.95 | 58.01 | 57.00 | 56.06 | 54.78 | 52.24 | 52.01 | 51.47 | -7.83 |
| iCaRL [4] | 68.68 | 52.65 | 48.61 | 44.16 | 36.62 | 29.52 | 27.83 | 26.26 | 24.01 | 23.89 | 21.16 | -39.92 |
| Rebalance [2] | 68.68 | 57.12 | 44.21 | 28.78 | 26.71 | 25.66 | 24.62 | 21.52 | 20.12 | 20.06 | 19.87 | -41.21 |
| TOPIC [6] | 68.68 | 62.49 | 54.81 | 49.99 | 45.25 | 41.40 | 38.35 | 35.36 | 32.22 | 28.31 | 26.28 | -34.80 |
| FSLL [3] | 72.77 | 69.33 | 65.51 | 62.66 | 61.10 | 58.65 | 57.78 | 57.26 | 55.59 | 55.39 | 54.21 | -6.87 |
| FSLL+SS [3] | 75.63 | 71.81 | 68.16 | 64.32 | 62.61 | 60.10 | 58.82 | 58.70 | 56.45 | 56.41 | 55.82 | -5.26 |
| **F2M** | 77.13 | 73.92 | **70.27** | **66.37** | **64.34** | **61.69** | **60.52** | **59.38** | **57.15** | **56.94** | **55.89** | **-3.41** |