# OpenReview forum: "Overcoming Catastrophic Forgetting in Incremental Few-Shot Learning by Finding Flat Minima"
_NeurIPS.cc/2021/Conference — NeurIPS 2021 Spotlight_

### Official Review · Reviewer_oBeE · 2021-07-07

**Rating:** 6
**Confidence:** 4

**Summary:**

This paper proposes an incremental few-shot learning method focusing on training the base model that can easily learn new classes, preserving the classification capability for the old classes. When training the base model, the proposed method, Finding Flat Minima (F2M) finds the flat minima where the loss function does not change a lot even if a random noise is injected. When the novel classes are given, F2M updates the feature extractor and clamp the parameters to make the feature extractor resides in the flat minima region.

**Limitations And Societal Impact:**

The authors did not addressed the limitations of  the paper.

**Main Review:**

[Strength]
1.	The idea of finding flat minima to overcome the catastrophic forgetting and ensure the base performance is interesting. The proposed F2M outperforms the other incremental few-shot learning methods for all sessions except for the base classes training phase (t=1).
2.	The proposed F2M does not saves the data from prior sessions except for prototypes for inference.
3.	The paper is well written and easy to follow. Also, the detailed ablation experiment results are provided to help understanding.

[Weakness]
1.	Although the paper argue that proposed method finds the flat minima, the analysis about flatness is missing. The loss used for training base model is the averaged loss for the noise injected models, and the authors provided convergence analysis on this loss. However, minimizing the averaged loss across the noise injected models does not ensure the flatness of the minima. So, to claim that the minima found by minimizing the loss in Eq (3), the analysis on the losses of the noise-injected models after training is required.
2.	In Eq (4), the class prototypes before and after injecting noise are utilized for prototype fixing regularization. However, this means that F2M have to compute the prototypes of the base class every time the noise is injected: M+1 times for each update. Considering the fact that there are many classes and many samples for the base classes, this prototype fixing is computationally inefficient. If I miss some details about the prototype fixing, please fix my misunderstanding in rebuttal.
3.	Analysis on the sampling times $M$ and noise bound value $b$ is missing. These values decide the flat area around the flat minima, and the performance would be affected by theses value. However, there is no analysis on $M$ and $b$ in the main paper nor the appendix. Moreover, the exact value $M$ used for the experiments is not reported.
4.	Comparison with single session incremental few-shot learning is missing. Like [42] in the main paper, there are some meta-learning based single session incremental FSL methods are being studied. Although this paper targets on multi-session incremental FSL with different setting and different dataset split, it would be more informative to compare the proposed F2M with that kind of methods, considering that the idea of finding flat minima seems valuable for the single session incremental few-shot learning task too.

* There is a typo in Table 2 – the miniImageNet task is 5-way, but it is written as 10-way.

-------------------------------------------------------------------------------------------------------------------------------------------
Post Rebuttal
- Reviewer clarified the confusing parts of the paper, and added useful analysis during rebuttal. Therefore, I raise my score to 6.


**Time Spent Reviewing:**

5

---

> ### Author Response · Authors · 2021-08-10
> **Response to Reviewer oBeE**
>
> Thank you for the constructive feedback! We are happy to respond to your comments and questions.
>
> **Q1: The analysis about flatness is missing. To claim that the minima found by minimizing the loss in Eq (3), the analysis on the losses of the noise-injected models after training is required.**
>
> **A1**: Thank you for the nice suggestion! To analyze the flatness of the found minima $\theta\^{\star}$, we sample the noise for 1000 times. For each time, we inject the sampled noise to $\theta\^{\star}$ and calculate the loss $\mathcal{L}\_i$. Then, we adopt the following indicator $I$ and variance to measure the flatness:
> $$
> I = \frac{1}{1000}\sum\_{i=1}^{1000}(\mathcal{L}\_i-\mathcal{L}\^{\star})\^2, ~~~~~~ \sigma\^{2} = \frac{1}{1000}\sum\_{i=1}^{1000}(\mathcal{L}\_i-\widehat{\mathcal{L}})\^2,
> $$
> where $\mathcal{L}\^\star$ denotes the loss of the minima $\theta\^{\star}$ and $\widehat{\mathcal{L}}$ denotes the average loss of $\\{\mathcal{L}\_i\\}\_{i=1}^{1000}$. Below we provide a comparison between the flat local minima found by our F2M and the sharp local minima obtained by the Baseline model in terms of flatness.
>
> **Indicator $I$**
>
> | Method/Dataset             |  Training set   | Testing set  |
> | :---                       |     :---:       |  :---:       |
> | Baseline                   |    0.2993       |   0.4582     |
> | **F2M**                    |  **0.0506**     | **0.0800**   |
>
>
> **Variance $\sigma\^{2}$**
>
> | Method/Dataset            |  Training set   | Testing set  |
> | :---                       |     :---:       |  :---:       |
> | Baseline                   |    0.1451       |   0.2395     |
> | **F2M**                    |  **0.0296**     | **0.0334**   |
>
> The results clearly demonstrate that our method can find a flat local minima. We will include the analysis in the revised manuscript.
>
>
> **Q2: In Eq (4), F2M needs to compute the prototypes of the base class every time the noise is injected: M+1 times for each update. Considering the fact that there are many classes and many samples for the base classes, this prototype fixing is computationally inefficient.**
>
> **A2**: Similar to SDC [56] and common practice, we use batch update. Specifically, for each update iteration, we sample a batch of data containing $N\_w$ classes and $N\_s$ samples per class to train the model with the loss defined in Eq (4). The class prototypes are computed/approximated using the batch data only. Therefore, the computation efficiency is not a significant issue.
>
> **Q3: Analysis on the sampling times M and the flat region bound b is missing.**
>
> **A3**: Empirically, we observe that the performance is quite stable for $M\in [2, 4]$. Since a larger $M$ may increase the training time, we set $M=2\sim 4$ in our experiments. Please refer to our response to Reviewer P52B for the analysis of $b$.
>
> **Q4: Comparison with single-session incremental few-shot learning methods such as [42] is missing. Although this paper targets on multi-session incremental FSL with different setting and different dataset split, it would be more informative to compare the proposed F2M with single-session incremental FSL methods, since the idea of finding flat minima seems valuable for this problem too.**
>
> **A4**: We did try to compare our F2M with [42]. However, we find it very difficult to make a fair comparison, because they are very different in terms of evaluation protocols ([42] is based on meta-learning, and hence the evaluation method and metric are totally different from ours and other baselines), data split ([42] uses 64 classes as base classes), and network structure ([42] adopts a modified version of ResNet-10 as the backbone network). A fair comparison requires re-implementing [42] under our setting or re-implementing our method under their setting, either of which needs a lot of effort.
>
> **Q5: There is a typo in Table 2 – the miniImageNet task is 5-way, but it is written as 10-way.**
>
> **A5**: Thanks for pointing this out!

---

> > ### Comment · Reviewer_oBeE · 2021-08-23
> > **Post-rebuttal**
> >
> > I appreciate that the authors provided the appropriate clarification and additional analysis.  Analysis on flatness of minima and flat region bound helps my understanding on the proposed method.
> >
> > Although my concern about computational inefficiency is not fully resolved, the authors provided the information about sampling times(M) which is quite small, I guess the additional computation would be reasonable.
> >
> > If the authors promise to add details about M and b to the paper (exactly like M=2, not like M= 2~4), I am willing to raise my score from 4 to 6.

---

> > > ### Author Response · Authors · 2021-08-23
> > > **Thank you**
> > >
> > > We thank the reviewer for accepting our response! For the results reported in the manuscript, we use $M=4$ for CIFAR-100 and *mini*ImageNet and $M=2$ for CUB-200-2011.
> > > We will definitely include these details, along with the analysis about $M$, $b$, and flatness in the revised version to make the work more complete and convincing. We will also release the source code to ensure reproducibility and facilitate future research. Thank you again for the valuable suggestion and for trusting us in further improving this work!

---

### Official Review · Reviewer_dLsD · 2021-07-15

**Rating:** 6
**Confidence:** 4

**Summary:**

The proposed method is a nice trial for tackling the issues in a challenging data setting --- class incremental learning with few-shot samples in every new class. The paper writing is clear to show that the converge of the proposed method is theoretically guaranteed and the performance shows better than some baselines --- incremental learning methods (but missing incremental few-shot learning methods introduced in Related Works [48, 42, 62, 9, 34, 61]).

**Main Review:**

I have two concerns about this paper. One is the performance should be highly related to the discrepancy gap between base classes and the coming (new) classes in incremental learning phases but there is no analysis on this. The other one is the motivation is not very strong to use in the only setting of few-shot samples in new classes. Please check the detailed comments.

1. The performance looks good but this paper only shows its comparisons with some incremental learning methods (baselines methods implemented in a new incremenral few-shot settings). The comparisons to closely related works such as [48, 42, 62, 9, 34, 61] are missing totally.

2. The performance of the models are related to the pattern differences between base classes and the coming (new) classes, which is not analyzed in the paper (as it used only one fixed setting of class splits). Given a larger difference between base and new classes, does the found flat minima (using base classes) still work for learning new classes? In addition, the experiments are a bit limited to small datasets.

3. Flat minima regions guarantee a good generalization of the trained model to new classes. Is this especially helpful for incremental learning in few-shot settings? It is interesting to see its empirical performance on the standard settings of class incremental learning, e.g., using ImageNet which is large-scale or at least using Sub-ImageNet.





**Time Spent Reviewing:**

4 hours

---

> ### Author Response · Authors · 2021-08-10
> **Response to Reviewer dLsD**
>
> Thank you for the constructive feedback! We are happy to respond to your comments and questions.
>
> **Q1:The comparisons to closely related works such as [48, 42, 62, 9, 34, 61] are missing totally.**
>
> **A1**: We did compare with the most recent incremental few-shot learning methods including [48], [34], and [8]. We apologize for causing the misunderstanding. Due to duplicated references, [48] and [34] are actually [49] and [35] in Table 1, 2, and 3.
>
> Despite our due diligence, the source codes of [9] and [62] are not available, and the code of [61] was released after the NeurIPS submission deadline. Although we tried to re-implement these methods, we were not able to determine some important details such as the number of network channels and the way of image transformation based on the information provided in the papers. [42] targets on single-session incremental few-shot learning, which is different from the multi-session problem setting considered in our paper. Moreover, the evaluation protocol, dataset/class split, and network structure of [42] are all very different from ours and other baselines, making it difficult to do a fair comparison.
>
> **Q2: The performance of the models are related to the pattern differences between base classes and the coming (new) classes, which is not analyzed in the paper (as it used only one fixed setting of class splits). Given a larger difference between base and new classes, does the found flat minima (using base classes) still work for learning new classes? In addition, the experiments are a bit limited to small datasets.**
>
> **A2**: CIFAR-100, *mini*ImageNet, and CUB-200-2011 are commonly used datasets for evaluating incremental few-shot learning methods, and we follow common practice for ease of comparison. Both CIFAR-100 and *mini*ImageNet have a large diversity of classes, and we split the classes randomly to ensure the generality of the results. To further demonstrate the performance of our model is statistically significant, below we report additional results on another two random class splits with different random seeds.
>
> **Classification accuracy on CIFAR-100 for 5-way 5-shot incremental learning (with random seed 1993 for class splitting).**
>
> | Method/Sessions            |      1         |    2     |   3     |   4      |   5      |   6      |   7    |    8     |    9    |
> | :---                       |     :---:      |  :---:   |  :---:  |   :---:  |   :---:  |  :---:   |  :---: |  :---:   |  :---:  |
> | Baseline                   |     67.38      |  63.29   |  60.41  |  57.66   |   54.68  |  52.42   |  49.66  |  47.58  |  45.78  |
> | ICaRL [41]                 |     65.80      |  58.95   |  55.76  |  52.78   |   49.71  |  47.56   |  44.48  |  42.60  |  40.90  |
> | Rebalance [20]             |   **69.55**    |  63.67   |  59.48  |  55.80   |   51.82  |  48.71   |  45.16  |  42.38  |  39.99  |
> | FSLL [34]                  |     67.38      |  57.05   |  54.50  |  53.21   |   52.02  |  50.22   |  48.00  |  46.02  |  44.19  |
> | **F2M**                    |     67.18      |**63.75** |**60.98**|**58.26** | **55.41**|**52.91** |**50.22**|**48.29**|**46.63**|
>
>
>
> **Classification accuracy on CIFAR-100 for 5-way 5-shot incremental learning (with random seed 11 for class splitting).**
>
> | Method/Sessions            |      1         |    2     |   3     |   4      |   5      |   6      |   7     |    8    |    9    |
> | :---                       |     :---:      |  :---:   |  :---:  |   :---:  |   :---:  |  :---:   |  :---:  |  :---:  |  :---:  |
> | Baseline                   |     67.77      |  64.14   |  60.13  |  57.10   |   54.34  |  51.47   |  49.67  |  47.68  |  45.58  |
> | ICaRL [41]                 |     66.27      |  59.09   |  55.02  |  51.88   |   49.11  |  46.26   |  44.55  |  42.57  |  40.22  |
> | Rebalance [20]             |   **68.92**    |  63.96   |  58.69  |  54.58   |   51.00  |  47.31   |  44.80  |  42.26  |  39.60  |
> | FSLL [34]                  |     67.77      |  59.11   |  56.27  |  54.52   |   52.65  |  49.60   |  48.01  |  46.39  |  44.22  |
> | **F2M**                    |     67.92      |**64.55** |**60.55**|**57.62** | **54.91**|**51.94** |**50.22**|**48.22**|**46.11**|
>
> Indeed, it would be interesting to investigate a more challenging problem setting, cross-domain incremental few-shot learning, which naturally assumes a large difference between base classes and new classes. Thanks for the nice question!
>
> **Q3: Flat minima regions guarantee a good generalization of the trained model to new classes. Is this especially helpful for incremental learning in few-shot settings? It is interesting to see its empirical performance on the standard settings of class incremental learning, e.g., using ImageNet which is large-scale or at least using Sub-ImageNet.**
>
> **A3**: Thanks for the nice suggestion! Our method is designed for incremental few-shot learning. To avoid excessive training and overfitting, it suffices to tune the model in a relatively small region. Nevertheless, our core idea can be potentially used for incremental learning. Previous IL methods (e.g., EWC, Rebalance, ICaRL) try to avoid forgetting in incremental learning sessions and tune the model around a sharp minima. It is possible to improve these methods by finding the flat local minima in the base training session. For example, we can search for a less flat but larger local minima region in the base training stage and tune the model within this region during incremental learning sessions, where previous techniques such as EWC can be used to constraint the model parameters. We also plan to explore this direction in the future.

---

> > ### Comment · Reviewer_dLsD · 2021-08-19
> > **after rebuttal**
> >
> > Thanks for the feedback.
> >
> > Regarding A1, thanks for clarifying the flaws in references. Please make the revision. For other related works, it was claimed a lot of difficulties which means authors did not solve the question during rebuttal. These might be done later but not sure.
> >
> > Regarding A2, I would not buy this. Actually, a simple trial can be done by explicitly controlling the splits of the training data, e.g., base classes restricted on animal classes and incremental phases on only vehicle classes (on CIFAR-100). New experiments of random splits (in the rebuttal texts) still get similar results as submitted ones, which therefore does not answer my question.
> >
> > Regarding A3, thanks for the explanation and mentioning example methods that can be incorporated in future work.
> >
> > Therefore, I keep my original rating.

---

> > > ### Author Response · Authors · 2021-08-19
> > > **2nd Response to Reviewer dLsD (Part A)**
> > >
> > > Thank you for taking the time to read our response and letting us know your concern! We want to further clarify some issues regarding your questions.
> > >
> > > Regarding A1 (missing baselines), **we have compared with as many recent methods as possible, and please note that multi-session IFL is a very new problem.**
> > > We have compared with [48]/[49] (CVPR 2020) (*the first paper* on multi-session IFL), [34]/[35] (AAAI 2021), and [8] (ICLR 2021). We did not compare with the rest because
> > >
> > > - [9] (CVPR 2021) - Too new, code not available.
> > >
> > > - [61] (CVPR 2021) - Too new, code released on June 22, *1 month after the NeurIPS submission deadline*.
> > >
> > > - [62] - A journal submission currently under review, code not available.
> > >
> > > - [42] (NeurIPS 2019) - It considers single-session IFL using meta-learning, with a very different setting and evaluation protocol from ours and other methods [48,34,8,9,61,62].
> > >
> > > Regarding A2 (class split), **we follow existing works [48,34,8,9,61,62] to use a fixed split for all the methods for a fair comparison and demonstrate the statistical significance of our results with multiple random class splits.** In our understanding, it is a standard practice (and a practical one) to use random splits. To further address your concern, we will follow your suggestion to conduct a trial on CIFAR-100 and provide the results soon. Thank you for your patience in advance!

---

> > > > ### Comment · Reviewer_dLsD · 2021-08-24
> > > > **Final comments**
> > > >
> > > > Thanks for the clarification and the additional experiments. Most of my questions are solved. I improve the rating to 6.
> > > >
> > > > Overall it is a good and concise approach to tackling challenging learning problems. Hope there can be some additional (or in future work) evaluation of its performance on more general tasks, which makes more impact potentially.

---

> > > > > ### Author Response · Authors · 2021-08-24
> > > > > **Thank you**
> > > > >
> > > > > We thank the reviewer for accepting our response and appreciate the valuable comments and suggestion! We are also very interested to explore whether our approach or its modified version can be applied for more general tasks such as class incremental learning as you suggested. Thank you again for your time and support for this manuscript!

---

> > > ### Author Response · Authors · 2021-08-21
> > > **2nd Response to Reviewer dLsD (Part B)**
> > >
> > > ### Additional experiments with different class splits on CIFAR-100
> > >
> > > The 100 classes in CIFAR-100 are grouped into 20 superclasses, and each superclass contains 5 classes. We split the classes according to the superclasses and provide results on two different class splits.
> > >
> > > **Case 1**: The base classes consist of *aquatic mammals, fish, insects, reptiles, small mammals, and large carnivores* (30 classes in total). The few-shot novel classes consist of *household furniture, vehicles2, flowers, and food containers* (20 classes in total).
> > >
> > > **Case 2**: The base classes consist of *aquatic mammals, fish, insects, reptiles, household furniture and small mammals* (30 classes in total). The few-shot novel classes consist of *people, vehicles2, flowers, food containers* (20 classes in total).
> > >
> > > **Classification accuracy for 5-way 5-shot incremental learning with class split case 1.**
> > >
> > > | Method/Sessions            |      1 (Animals)        |    2 (Vehicles2)     |      3 (Flowers)        |    4 (Food Containers)    |    5 (Household Furniture)     |
> > > | :---                       |     :---:      |  :---:   |      :---:     |   :---:  |  :---:  |
> > > | Baseline                   |     63.07      |  56.32   |     51.40      |  46.85   |  43.55  |
> > > | ICaRL [41]                 |   **63.30**    |  55.10   |     49.12      |  44.46    |  40.95  |
> > > | Rebalance [20]             |     63.03      |  52.06   |     45.87      |  39.35   |  35.24  |
> > > | FSLL [34]                  |     63.07      |  50.72   |     44.53      |  40.73   |  38.00  |
> > > | **F2M**                    |     62.53      |**56.63** |  **51.87**     |**47.54** |**44.10**|
> > >
> > >
> > >
> > > **Classification accuracy for 5-way 5-shot incremental learning with class split case 2.**
> > >
> > > | Method/Sessions            |      1 (Animals+Furniture)       |    2 (People)   |      3 (Vehicles2)      |    4 (Flowers)  |    5 (Food Containers)  |
> > > | :---                       |     :---:      |  :---:   |      :---:     |   :---:  |  :---:  |
> > > | Baseline                   |     63.07      |  54.30   |     50.16      |  46.19   |  43.16  |
> > > | ICaRL [41]                 |     62.57      |  51.67   |     47.51      |  42.98   |  39.63  |
> > > | Rebalance [20]             |   **63.50**    |  49.62   |     44.67      |  39.68   |  35.64  |
> > > | FSLL [34]                  |     63.07      |  49.45   |     46.30      |  41.94   |  39.33  |
> > > | **F2M**                    |     62.87      |**54.82** |  **50.88**     |**46.88** |**43.83**|
> > >
> > >
> > > The results show that even with a large difference between the base classes and novel classes, our F2M still consistently outperforms other methods, indicating its robustness and effectiveness. We appreciate the reviewer pointing out this issue, which helps us further evaluate the robustness of our method and look deeper into the IFL problem. Please let us know if you have additional concerns or questions, and we would be happy to discuss them.

---

### Official Review · Reviewer_aQXy · 2021-07-16

**Rating:** 7
**Confidence:** 4

**Summary:**

The paper introduces a novel approach for few-shot incremental learning, which searches for the flat local minima of the base training objective function and then fine-tune the model parameters within the flat region on new tasks. This way, the model could learn the new classes without forgetting the old ones. The approach is well-motivated and is scientifically sound.

**Ethical Concerns:**

Not applicable.

**Limitations And Societal Impact:**

The paper does not have a negative societal impact.

**Main Review:**

The paper is well-documented and clearly written. The related work section covers the most relevant papers in the literature.  Extensive experiments have been performed and the approach was conveniently compared with the current state-of-the-art, demonstrating its superiority.
Here are my concerns:
1. Section 4.1: Definition 1, Condition 1: 'b_i' is not introduced. Who is 'epsilon'?
2. In Section 4.3, you make a series of assumptions. How realistic are they in practice? What measures do you take in order to guarantee that they are met?
3. How do you find the optimal parameter 'b'? What value do you use in your experiments? A study of the effect of the parameter 'b' on the performance would be interesting.


**Time Spent Reviewing:**

3 hours reading + 1 hour filling the questionnaire

---

> ### Author Response · Authors · 2021-08-10
> **Response to Reviewer aQXy**
>
> Thank you for the positive and constructive feedback! We are happy to respond to your comments and questions.
>
> **Q1: What are $\mathbf{b}_i$ and $\epsilon$ in Condition 1 of Definition 1?**
>
> **A1**: $\mathbf{b}$ is a constant vector with each component $\mathbf{b}\_i=b$ $(i= 1,2,\ldots)$. $\epsilon$ is a vector that satisfies $-\mathbf{b} \preceq\epsilon\preceq \mathbf{b}$, i.e., $-b \leq \epsilon\_i \leq b$, $\forall i$. Condition 1 of Definition 1 defines a flat region with bound $b$, that is, for any $\theta\^\star - \mathbf{b} \preceq \theta \preceq \theta\^\star + \mathbf{b}$, it satisfies that $\mathcal{L}(z;\theta) = \mathcal{L}(z;\theta\^\star)$.
>
> **Q2: How realistic are the assumptions in Section 4.3 in practice? What measures do you take in order to guarantee that they are met?**
>
> **A2**: Our convergence analysis mainly adopts the common assumptions used in the convergence analysis of gradient-based optimization algorithms such as SGD [3, 27], which are realistic and can be easily met in practice.  **Assumption 4.1** is a natural and widely-adopted assumption, which ensures the gradient does not change arbitrarily quickly w.r.t. the parameter vector, otherwise it cannot be a good indicator for how far to move to decrease the loss function $R$. For **Assumption 4.2**, the first condition assumes that the loss $R$ is bounded by a minimum value $R\^{*}$, which is natural and practical. The second condition is a strict assumption which is made to simplify the proof, but it can be easily relaxed to a general and easily-met condition that there exist $\mu\_1 \geq \mu\_2>0$  satisfying $
> \||\mathbb{E}\_{z\_{k},\epsilon\_j}\[g(z\_k;\phi\_k +\epsilon\_j, \psi\_k)\]\||\_2 \leq \mu\_1\||\nabla R(\theta\_k)\||\_2 \quad$ and $\nabla R(\theta\_k)^{T}\mathbb{E}\_{z\_{k},\epsilon\_j}\[g(z\_k;\phi\_k +\epsilon\_j, \psi\_k)\] \geq \mu\_2\||\nabla R(\theta\_k)\||\_2\^2
> $. Under this condition, the convergence can be proved in a similar way using the techniques presented in the Appendix. The third condition assumes that the variance of gradients cannot be arbitrarily large, which is also reasonable in practice. **Assumption 4.3** can be easily met, since in practice the learning rate $\alpha\_k$ is usually far less than $1$ and decreasing w.r.t. $k$.
>
> **Q3: How to find the optimal parameter $b$, and what value is used in the experiments? A study of the effect of $b$ on the performance would be interesting.**
>
> **A3**: Thanks for the nice suggestion! In our experiments, we choose $b$ by validation. Please see our response to Reviewer P52B for an ablation study on $b$.

---

> > ### Comment · Reviewer_aQXy · 2021-09-02
> > **Final decision**
> >
> > The authors properly addressed all the raised questions. Therefore, I maintain my initial rating which is 7,

---

### Official Review · Reviewer_P52B · 2021-07-17

**Rating:** 9
**Confidence:** 4

**Summary:**

This paper describes an approach to incremental few-shot learning based on
finding a good starting configuration in parameter space that leads to more
stable incremental few-shot learning with less forgetting. The authors propose
to learn an approximately flat region on the initial training set and to then
constrain few-shot incremental learning to remain within this flat region and
thus minimize forgetting. Experimental results are given on a wide range of
few-shot incremental benchmarks, including CIFAR-100, miniImageNet, and CUB-200.


**Limitations And Societal Impact:**

Discussion of limitations and societal impact is entirely missing.

**Main Review:**

This paper is very well-written with excellent motivations, empirical and
theoretical follow-through, and convincing experimental evaluation. I have
picked and poked and prodded at the work, but can find very little to improve. I
especially appreciated the empirical motivations of the work in section 3, as
well as the convergence analysis. A few that I think should be addressed:

1. The motivation of starting from a flat region and regularizing incremental
   learning based on it is reminiscent of Elastic Weight Consolidation, but
   motivated from a different starting point -- pun intended since EWC does not
   constrain the initial starting point, but rather only regularizes based on an
   approximation of the loss surface at an initial solution. It would be
   interesting to see how and why regularization using the Laplace approximation
   of the loss landscape (a la EWC) around a flat minima works when compared to
   clamping to remain in the flat region.
2. I do not find any discussion of the flat region bound $b$ used to determine
   the flat region. I expect a smaller $b$ leads to a smaller flat region and
   thus less plasticity, and a larger $b$ to a larger flat region and more risk
   of forgetting. It would be nice to see an ablation on this parameter to
   understand better this tradeoff.

**To summarize**: I really like this paper and I think it is novel approach that
adds something very interesting to the discussion on incremental learning.

**Post-rebuttal Summary**: I appreciate the author efforts to clarify and provide additional ablations. Other reviewers point to somewhat more critical blemishes and/or questions about the proposed approach, but I am still very positive about the work. The manuscript is very well-reasoned and well-written, and I think it adds something interesting to the discussion on few-shot IL. I do suggest that the authors consider carefully the comments from reviewer dLsD on carefully articulating the contribution of the work and the relationship between flat minima, few-shot learning, and incremental learning.

**Time Spent Reviewing:**

2.5

---

> ### Author Response · Authors · 2021-08-10
> **Response to Reviewer P52B**
>
> Thank you for the encouraging and insightful comments! We are happy to respond to your comments and questions.
>
> Indeed, we think a la EWC can potentially work better than simple clamping for regularization around a flat local minima, and we plan to explore this direction as a way of further improving the performance of incremental few-shot learning in the near future. Thank you for the nice suggestion!
>
> **Ablation study of the flat region bound $b$**
>
> We conduct an empirical study of the flat region bound $b$ for 5-way 5-shot incremental learning on CIFAR-100. We report the test accuracy in session 1 (base session) and session 9 (last session) w.r.t. varying $b$. The top 3 results in each row are in boldface.
>
> |   $b$                         |       0.0025   |    0.005 |   0.01  |   0.02   |   0.04   |   0.08   |
> | :---                       |     :---:      |  :---:   |  :---:  |   :---:  |   :---:  |  :---:   |
> | session 1 (60 base classes)                 |   **64.85**    |   64.67  |  **64.81**  |  **64.71**   |   63.30  |  62.25   |
> | session 9 (100 classes)                 |     44.16      |   **44.54**  |  **44.58**  |**44.67** |   43.75  |  43.04   |
> | session 9 (60 base classes)  |     **59.58**      |   **59.69**  |**59.73**|  59.44   |   58.38  |  57.21   |
> | session 9 (40 novel classes) |     21.03      |   **21.81**  |  **21.86**  |**22.52** |   21.80  |  21.77   |
>
> It can be seen that the best results are achieved for $b\in[0.005, 0.02]$. A larger $b$ (e.g., 0.04 or 0.08) leads to a significant performance drop on base classes, even in the base session 1, which indicates that a large flat region around a good local minima may not exist. Meanwhile, a smaller $b$ (e.g., 0.0025) results in a performance decline on novel classes, which is due to the less plasticity of the small flat region, exactly as you expected. The study illustrates the trade-off effect of $b$. In our experiments, we choose $b$ by validation. We will include the results and analysis in the revised manuscript.

---

### Author Response · Authors · 2021-08-10
**Summary of Rebuttal**

We thank all the reviewers for taking their valuable time in reviewing our manuscript! We really appreciate the helpful comments and suggestion towards improving our work. Below is a summary of our response to the comments and questions.

- We have clarified a misunderstanding in the comments of Reviewer dLsD about missing baselines, which was caused by our mistake in duplicating the references (sorry!).

- To address the common question of Reviewer P52B, Reviewer aQXy, and Reviewer oBeE about the flat region bound $b$, we have provided an ablation study of the effect of $b$ in our response to Reviewer P52B.

- To address the question of Reviewer dLsD about the effect of class split, we have provided additional experimental results in our response to Reviewer dLsD.

- To address the question of Reviewer oBeE about the flatness of the local minima found by our method, we have provided an analysis in our response to Reviewer oBeE.

---

### Decision · Program_Chairs · 2021-09-27

**Decision:**

Accept (Spotlight)

**Comment:**

After reading the paper and other reviews, and interaction with the authors via the rebuttal, all reviewers recommend to accept the paper. It's well written and motivated, and the proposed method, which is relatively simple, is reasonably evaluated in the context of incremental few-shot learning.